# Multi-Faceted Analysis of Systematic Risk-Based Wind Energy Investment Decisions in E7 Economies Using Modified Hybrid Modeling with IT2 Fuzzy Sets

**Dejun Qiu [1],\*, Hasan Dinçer [2] , Serhat Yüksel [2],\* and Gözde Gülseven Ubay [2]**

[1] Learning Achievement Certification Centre, Jilin Radio and TV University, Changchun 130022, China
[2] School of Business, Istanbul Medipol University, Kavacık South Campus, 34810 Istanbul, Turkey; hdincer@medipol.edu.tr (H.D.); gozde.ubay@std.medipol.edu.tr (G.G.U.)
\* Correspondence: qiudj317059@sina.com (D.Q.); serhatyuksel@medipol.edu.tr (S.Y.)

**Abstract:** This study aimed to analyze the systematic risks of wind energy investments. Within this framework, E7 countries are included in the scope of the examination. A large literature review was carried out and 12 different systematic risk factors that could exist in wind energy investments were identified. The analysis process of the study consisted of two different stages. First, the specified risk criteria were weighted with the help of the interval type 2 (IT2) fuzzy decision-making trial and evaluation laboratory (DEMATEL) method. Second, E7 countries were ranked according to the risk management effectiveness in wind energy investments. In this process, the IT2 fuzzy Vlsekriterijumska Optimizacija I Kompromisno Resenje (VIKOR) approach was taken into consideration. The findings show that volatility in exchange rates and interest rates were the most important risks in wind energy investments. In addition, it was determined that China and Indonesia were the most successful countries in managing risks in wind energy investments. In contrast, India, Russia, and Turkey were determined to be the least successful. Additionally, the IT2 fuzzy technique for order preference by similarity to ideal solution (TOPSIS) method was applied as a robustness check of the extended VIKOR method. It was concluded that the ranking results of the IT2 fuzzy TOPSIS method were similar to the results of the IT2 fuzzy VIKOR. It can be understood that the proposed ranking method was consistent with the comparative analysis results. From this point of view, it was observed that countries should take measures regarding their exchange rate and interest rate risks in order to increase the efficiency in wind energy investments. In this context, companies should first ensure that they do not have a foreign exchange short position in their balance sheets by conducting an effective financial analysis. In addition, it is important to use financial derivatives to minimize the exchange rate and interest rate risks. Using these results, it will be possible to manage this risk by taking the reverse position for the existing foreign currency and interest risk. In this way, it will be possible to increase the efficiency of wind energy investments, which will contribute to the social and economic development of each respective country.

**Keywords:** wind energy; investment decision; IT2 fuzzy DEMATEL; IT2 fuzzy VIKOR; IT2 fuzzy TOPSIS

## 1. Introduction

Energy helps to meet the basic needs of people, such as providing warmth and lighting. In addition, energy is also the most important raw material in industrial production. Considering these issues, it can be understood that energy is vital for a country. In other words, providing energy for both the social and economic development of countries is essential. In this context, countries that have

their own energy resources can benefit from these resources [1]. This situation gives these countries a serious cost advantage. On the other hand, countries that do not have sufficient energy resources have to supply the energy they need from outside, which makes these countries dependent on energy. Because of this problem, countries are developing several strategies to produce their own energy [2].

Renewable energy sources are an important alternative that allows countries to produce their own energy. The most important feature of these energy alternatives, which come in different forms, such as wind, solar, biomass, geothermal, and hydroelectricity, is that they obtain their resources from nature. Therefore, the sources of these energy alternatives are not exhausted [3]. In this way, countries reduce their external dependence on energy by producing their own energy. Thus, energy imports will be lower and the current account balance will be positively affected. This will contribute positively to the economic development of the country. Another advantage of renewable energy sources is that they do not pollute the environment like fossil fuels. With these energy alternatives, there is no carbon gas emission to the atmosphere [4]. This is another reason for preferring renewable energy sources.

Wind energy can be expressed as motion energy obtained from the airflow that is the wind. Wind turbines are set up to convert this kinetic energy in the wind into mechanical energy and then into electrical energy [5]. Hence, it is possible to say that wind energy is an inexhaustible source because its raw material is wind. On the other hand, wind energy has some advantages over other renewable energy types. Since the wind is available at all times during the day, it is possible to obtain electricity continuously with the help of wind turbines. In contrast, regarding solar energy, electricity cannot be obtained after sunset [6]. In addition, wind energy does not take up much space like other types of renewable energy, which is one of the important advantages of this energy type.

Considering these issues, it can be understood that wind energy investments are very important for the social and economic development of the country. Thanks to these investments, it is possible to obtain energy that does not cause environmental pollution, as well as avoiding dependency on energy on the outside [7]. Therefore, it is important to develop strategies for countries to increase their wind energy investments. In this context, practices such as tax advantages and location support provided by the government will contribute to the motivation of the investors in this field. Another important issue in this process is that wind energy investors should effectively manage their risks. These investors face many different risks, internal or external, due to their activities. In this context, it is obvious that the most important risks are systematic risks [8]. The main reason for this type of investment is that they include the risks that occur independently of the company and which the company cannot prevent in its formation. In other words, wind energy investors can only take measures to manage these risks; they cannot prevent these risks from occurring.

Vulnerability in the macroeconomic conditions can create some systematic risks for wind energy investors, such as exchange rate risk. Some products in the wind turbine mechanism can be imported from abroad. Considering that these products are paid for in a foreign currency, an increase in the exchange rate will automatically increase the costs incurred by the company [9]. Another type of risk that is important in this process is high inflation. Costs incurred during the distribution of all purchased products and electricity generated as a result of wind energy will increase even more in a high inflation environment [10]. This will cause uncertainty in investments to increase. Interest rates are another type of risk that affects wind energy investors. In case the interest rate increases, the wind power company will have difficulties in finding loans for its new investments [11]. This will cause the company to experience financial problems.

In addition, some socioeconomic issues also lead to systematic risks in wind energy investments. Within this context, in the case of political problems in the country, investors are reluctant to make new investments. Moreover, as a result of political instability, different strategies can be implemented by the country's administration, which causes wind energy investors to experience some problems [12]. Another factor that plays an important role in this process is the social turmoil in the country. In a country where there is social conflict, the crime rate is likely to increase. This situation may cause wind energy investments to be negatively affected [13]. Rapid changes in the legal system can also reduce

the effectiveness of wind energy projects [14]. For example, a legal change in a country, such as a tax increase, can reduce the profitability of wind energy investments.

The systematic risks faced by wind energy investors may also arise from the market. In this process, one of these types of risks to be considered is the decrease in market demand. As an example, in a country where the population is decreasing or where an economic recession is experienced, the demand for electricity will decrease [15]. In this case, it will cause wind energy companies to decrease their profitability. In addition, the increased cost due to factors not connected to the company will also decrease the performance of the investments. For example, when taxes increase, raw materials will become more expensive and this will negatively affect the profitability of wind energy companies [16]. Another example of systematic risks is increased competition in the industry. This competitive environment will also lead to wind energy companies not being flexible about the price [17]. Consequently, the profitability of companies will be negatively affected by this process.

Furthermore, environmental factors also increase the systematic risks of wind energy companies. In this context, company investments may be adversely affected in a country where natural disasters occur [18]. The main reason for this is the possibility of these disasters physically damaging wind energy investments. In addition, in the case of problems, such as global warming, serious climate changes occur in the country [19]. This will affect wind energy investments negatively. For example, the efficiency of wind energy investments to be made in a country that does not receive enough wind will decrease. Building development in a region may also have an impact on the performance of wind energy investments. For instance, if there are very high-rise buildings in a region, this may lead to the wind power plants in that region not getting enough wind.

In this study, systematic risks in wind energy investments were analyzed. Within this framework, first, as a result of the literature review, 12 different risk types were determined for wind energy investments, which were sorted into 4 groups. After that, an investigation was carried out using the interval type 2 (IT2) fuzzy DEMATEL method to understand which of these risks were prioritized. Therefore, it was possible to find the answer to the question of which systematic risks were faced most by wind energy investors. Following these analyses, the E7 countries (Brazil, China, India, Indonesia, Mexico, Russia, Turkey) were ranked in terms of their risk situations regarding wind energy investments. These countries have developing economies, but they aim to reach the status of developed economies. Toward this aim, these countries try to implement many significant strategies in various areas. The main problem in this process is that these countries can sometimes take high risks to achieve this objective [20]. Thus, it is thought that risk analysis studies are very helpful for these countries to reach sustainable economic improvement.

In the analysis in this process, the IT2 fuzzy VIKOR model was taken into account. Moreover, the IT2 fuzzy TOPSIS method was also considered as a robustness check of the extended VIKOR method. According to the results of the analysis, it was revealed which countries were more successful at managing these risks. The main advantage of considering IT2 fuzzy sets in comparison with IT1 fuzzy sets is that IT2 fuzzy sets handle uncertainties better. In addition, many control designs can also be implemented in IT2 fuzzy sets [21]. Moreover, another important advantage of IT2 fuzzy sets is its relatively high flexibility and robustness in comparison with IT1 fuzzy logic [22].

It is possible to mention many different advantages of this study. First, by presenting the weighted systematic risks, it was possible to determine which risks the wind energy investors primarily faced. Consequently, the most important risks, which wind energy investors cannot prevent but can take precautions against were identified. This situation guides wind energy investors and academicians in their work. This may contribute to increasing wind energy investments in a given country. On the other hand, another important contribution of the study is the ranking of the E7 countries in terms of risk situations in wind energy investments. In this way, it will be possible to offer suggestions for these countries to better manage the process. E7 countries aim to become developed countries; to achieve this purpose, they try to increase their investment. Additionally, another significant point is that they can take too many risks when increasing their investments. Therefore, identifying important

systematic risks can help these companies to minimize these problems. Moreover, energy dependency is an essential economic problem for these companies. Hence, wind energy investments have a positive influence when it comes to solving this problem.

This study includes five different sections. In the first part of the study, information about the importance of energy, the benefits of renewable energy, the advantages of wind energy, and the risks faced by wind energy investors are shared. In the second part of the study, the results of a comprehensive literature review on the systematic risks encountered in wind energy investments is given. The third part of the study covers the theoretical knowledge of IT2 fuzzy DEMATEL and IT2 fuzzy VIKOR methods used in the analysis process. In the fourth part of the study, the results of the analysis are included. The last part of the study focuses on the strategy proposals that were developed based on the analysis results obtained.

## 2. Literature Review

Many studies have been conducted in the literature on renewable energy, especially wind energy. The vast majority of these studies focused on wind energy investments since it is a growing and preferred sector in many countries. In wind energy investments, besides other aspects, risk factors are an important topic that must be examined. In the literature, many studies examine these factors to pave the way for investors because most investors have to act professionally in these investments that require high technology.

### 2.1. Literature on the Risk Factors of Wind Energy Investments

Wind energy investments are complex projects that require professional engineering knowledge. In this sector, many different specific products are used and supplied as raw materials. Since these products are specific, they can be imported from abroad. When this occurs, the payment is made in a foreign currency. Therefore, an increase in the exchange rate will automatically increase the costs incurred by the company. In the literature, as Keeley and Matsumoto [5] mentioned, for risk-averse investors, high volatility in the currency exchange rate may be a deterrent since the volatility can be regarded as an additional cost. More importantly, wind energy companies can get their loans in a foreign currency because of their low cost. However, the payment of these loans will be in a foreign currency. Therefore, an increase in the exchange rate will increase the amount of the repayments. Even if companies buy the products with a foreign exchange, their profit will be in the local currency [3]. In these wind energy investments, which involve quite a long process, a sudden change of the exchange rate is a risk that cannot be ignored by investors [9].

Another risk that should not be ignored is the inflation rate in a country. Inflation is a serious problem for wind energy projects. The initial cost of these projects is very high. Therefore, the project can become even more costly when inflation increases [10]. Costs incurred during the distribution of all products purchased and electricity generated as a result of wind energy will increase in a high-inflation environment. Another problem brought about by the increase in inflation is related to operational costs. When inflation increases, it will cause the workers' salaries to rise. This situation causes the costs of the company to increase. When the initial and operational costs rise, the effectiveness and profitability of the project will be negatively affected [20]. As Adaramola et al. [1] studied, as inflation increases, the return on investment decreases. Furthermore, when the interest rate increases, the wind power company will have difficulties in finding loans for its new investments. Wind energy projects are long-term projects, and companies may periodically need credit. It is possible that if the interest rate rises, the company may not find any new loans for the project that has started. This may cause unfavorable results [11]. Besides this, the increasing interest rate will harm the customers and suppliers of the company. This will cause the wind power company to experience difficulties in the supply chain [6].

Political problems are also one of the important factors that pose a risk for such large investments. In the case of political problems in the country, investors are reluctant to make new investments [4].

This is also important for wind energy investments with high initial costs and complicated processes. A change in power in a country can lead to different governance strategies and these changing strategies may have some negative effects on wind energy investors. Furthermore, there may be distortions in the previously mentioned macroeconomic factors as investors will be reluctant to invest in the absence of political stability. For wind energy to be sustainable, political stability must be ensured. Otherwise, there is a risk that the profitability of wind energy investments will decrease [5]. Political instability can trigger the deterioration of the peaceful environment in the society, which is concerning given that another risk factor for wind energy investors is social risk [12].

The country's social turmoil makes investors uneasy. Because these projects are high-cost and long-term investments, social cohesion is even more important for wind energy investors. They need to take the concerns and requirements of society into account [13]. In a country where there is a social conflict, the crime rate is likely to be high. If these problems cannot be controlled, there is a risk that wind energy investments will fail [7]. Deficiencies or problems in the legal system can affect wind energy investment projects [23,24]. As Elsner and Suarez [25] studied, in the case of an insufficient legal infrastructure, investors will experience uneasiness. The main reason for this is that there is a risk of a possible problem not being solved. Wind energy investors prefer to feel safe. Therefore, they want the legal infrastructure to be in place and sound. When this is not the case, wind energy investors will not want to invest in a country where the legal system is constantly and radically changed by political power. One of the most important factors affecting investors' decisions is the uncertainty situation in a country [14].

Wind energy investment companies initially have very high costs. However, some investors prefer to enter this risky investment as they expect to have high profitability rates in the long term. One of the important issues here is the market demand for electricity being produced [15]. This demand can be affected by many different factors. For example, in a country where the population is decreasing or where an economic recession is experienced, the demand for electricity will decrease. In this case, it will cause wind energy companies to decrease their profitability [26]. Furthermore, if the country has many fossil fuels, the demand for renewable energies may not be as high because of the cheap prices of fossil sources. In order to minimize this risk, investors must examine the market correctly and be open to market signals [27].

In order for wind energy investments to be profitable, the sales volume must be high and the operating costs must be low. Increasing costs due to factors not connected to the company in this process will decrease the performance of the investments [28]. For example, in an environment where raw materials become more expensive and taxes increase, there will be serious increases in cost items. In this case, it will cause the profitability of wind energy investments to decrease. Energy is an essential requirement for a country. Therefore, countries have to supply these energies continuously. As a result, this significantly attracts the attention of energy companies and many companies enter the sector aggressively [16]. In this case, it will cause the competition in the sector to increase. This competition will cause wind energy companies to be flexible in terms of price, and in this case, it will cause the mentioned companies to decrease their profitability [17]. Furthermore, in a country where natural disasters occur, company investments can also be negatively affected. It is a risk that investors should be careful of. The main reason for this is that these disasters physically damage the investments [29]. Wind energy investments are investments that can be made in a certain region and are quite tall. Therefore, it faces many natural disaster risks. As an example, natural disasters, such as storms, floods, and earthquakes may cause serious damage to these projects [18].

The main event required for wind energy investments is the presence of wind. In other words, in these investments, the wind must blow regularly in order to provide electricity continuously. In situations where problems, such as global warming, are experienced, serious climate changes will occur [30]. This will negatively affect wind energy investments. On the other hand, the climate conditions of the country are also important for the continuity of these investments. For example, wind energy investments in a country that do not receive wind may not be very effective. For this

reason, geological risks are very important for investors [19]. Besides natural geographical factors, there can be some geographical risks arising from humankind. Structuring plans in a region can have an impact on the performance of wind energy investments [31]. For example, if there are high-rise buildings in an area, this may lead to the wind power plants in that area not getting enough wind [32]. Therefore, the efficiency of the mentioned investments will decrease. As a result, in order for wind energy investments to be successful, the construction in the country should be well planned and this situation should be clarified using laws. The effectiveness of wind energy investments is always at risk in an area with distorted construction or at risk of being distorted [8].

### 2.2. Literature Review on Methodology

One of the important contributions of this study to the literature is methodological. IT2 fuzzy decision-making trial and evaluation laboratory (DEMATEL) and IT2 fuzzy VIKOR approaches were considered for the first time in this study regarding the risks in wind energy investments. Furthermore, the IT2 fuzzy TOPSIS method was also used as a robustness check. This situation increased the methodological originality of the study. The biggest advantage of the DEMATEL model over other similar methods is that it also shows a causality relationship between the variables. Moreover, group utility maximization and individual regret minimization can be considered in VIKOR approach, which was accepted as the main advantage of this method. Furthermore, with the help of the IT2 fuzzy methodology, uncertainties in the decision-making process could be minimized as well.

In this study, the effectiveness of risk management in wind energy investments in E7 countries was examined. For this purpose, IT2 fuzzy DEMATEL was considered to find the significance levels of the systematic risk criteria. This methodology has frequently been considered in the literature, especially in recent years. For instance, Dinçer et al. [33] conducted an analysis of the Turkish banking sector to evaluate the customer satisfaction of mobile phone applications. Pandey et al. [34] also made a similar analysis for different country groups. Additionally, Zhou et al. [35] tried to find the issues that finance companies should consider in order to finance large-scale energy investments. Dinçer and Yüksel [36] used the IT2 fuzzy DEMATEL approach to find the key strategies for tourism investments to be innovative. Similarly, Agrawal and Kant [37] aimed to determine the criteria for an efficient supplier selection using this approach. Furthermore, Dinçer et al. [38] focused on the performance measurement of international companies in the Baltic countries. Similarly, Toklu and Taşkın [39] aimed to measure the performance of SMEs with the help of this approach.

In the second stage of the analysis in this study, the E7 countries were ranked according to their effectiveness at wind energy investments. For this situation, the IT2 fuzzy VIKOR methodology was considered. This approach was preferred by researchers for various reasons. As an example, Wu et al. [40] aimed to choose a suitable supplier by using this methodology. In addition to this study, Efe [41] tried to select the best enterprise resource planning (ERP) software. Similarly, Guo et al. [42] considered the IT2 fuzzy VIKOR method when aiming to reduce greenhouse gas emissions by determining the risk assessment of $CO_2$ transmission pipelines. Moreover, Salimi et al. [43] examined the role of advertising on water consumption. Furthermore, Haider et al. [44] analyzed ways to prevent the contamination of limited water in dry and arid places. Additionally, Solangi et al. [45] focused on the electricity generation in Pakistan by considering the IT2 fuzzy VIKOR methodology. Furthermore, a decision-making scheme was prepared by Mete et al. [46] using the fuzzy VIKOR method for risk factors in the construction process of a natural gas pipeline.

In addition, the IT2 fuzzy TOPSIS method was applied as a robustness check of the extended VIKOR method in order to rank the E7 countries. The fuzzy TOPSIS methodology is also very popular in the literature. Within this scope, Chou et al. [47], Cavallaro et al. [48], and Efe [49] used this approach to evaluate technological development. Similarly, the subject of supplier selection has also been frequently considered by researchers. For instance, Memari et al. [50], Rashidi and Cullinane [51], and Bera et al. [52] aimed to find the best supplier using the fuzzy TOPSIS methodology. However, it was also identified that the number of studies using IT2 fuzzy TOPSIS are very limited in the

literature. In this framework, Yang et al. [53], Qin and Liu [54], and Yücesan et al. [55] are example studies in which this method was taken into account.

### 2.3. Literature Review Results

As a result of the literature review, it was established that the risk management issue in wind energy projects is very popular. These studies generally examined which risk factor was more effective on these investments. In this context, factors such as volatility in the exchange rates, interest rate risk, high inflation, political factors, legal risks, natural disasters, and competition in the market have been brought to the fore in general. As can be seen from this, it has been argued that different factors are underlined in the studies. Therefore, a new study is needed to examine which is more important for wind energy investments by taking into account all these factors at the same time. As a result of a detailed literature review, 12 systematic risk types for wind energy investments were determined in this study, and the analysis aimed toward determining which of these factors was most important. Another limitation was found as a result of the literature review, which relates to the analysis methods used. In these studies, it is seen that methods such as regression, analytic hierarchy process (AHP), and surveys are generally preferred. In this study, the IT2 fuzzy DEMATEL, IT2 fuzzy VIKOR, and IT2 fuzzy TOPSIS methods were used for the first time in the analysis of systematic risks in wind energy investments. This increases the methodological originality of this study.

## 3. Methodology

In this study, the systematic risks of wind energy companies were analyzed. In this context, the identified systematic risk types were primarily weighted using IT2 fuzzy DEMATEL. After that, the E7 countries were ranked using IT2 fuzzy VIKOR in terms of their success at managing the related risks. Furthermore, the IT2 fuzzy TOPSIS method was applied as a robustness check. Details of this innovative method mentioned is explained in this section.

### 3.1. IT2 Fuzzy DEMATEL

The DEMATEL method was used to determine which were the most important criteria that affected wind power investment. The most important advantage of this approach compared to others is that the effect-relationship analysis between variables can be made. In other words, in the DEMATEL method, it can be understood which variables affect others. It can be seen that this approach can also be considered with IT2 fuzzy sets [56]. For this purpose, first, decision-makers make evaluations and they are converted into the fuzzy sets. Second, an initial direct relation matrix ($\widetilde{Z}$) is created, as given in Equation (1):

$$\widetilde{Z} = \begin{bmatrix} 0 & \widetilde{z}_{12} & \cdots & \cdots & \widetilde{z}_{1n} \\ \widetilde{z}_{21} & 0 & \cdots & \cdots & \widetilde{z}_{2n} \\ \vdots & \vdots & \ddots & \cdots & \cdots \\ \vdots & \vdots & \vdots & \ddots & \vdots \\ \widetilde{z}_{n1} & \widetilde{z}_{n2} & \cdots & \cdots & 0 \end{bmatrix}. \tag{1}$$

In this process, the average values of these evaluations were considered, and the details are shown in Equation (2) [57]:

$$\widetilde{Z} = \frac{\widetilde{Z}^1 + \widetilde{Z}^2 + \widetilde{Z}^3 + \ldots \widetilde{Z}^n}{n}. \tag{2}$$

After that, a normalization process occurs using Equations (3)–(5):

$$
\widetilde{X} = \begin{bmatrix} \widetilde{x}_{11} & \widetilde{x}_{12} & \cdots & \cdots & \widetilde{x}_{1n} \\ \widetilde{x}_{21} & \widetilde{x}_{22} & \cdots & \cdots & \widetilde{x}_{2n} \\ \vdots & \vdots & \ddots & \cdots & \cdots \\ \vdots & \vdots & \vdots & \ddots & \vdots \\ \widetilde{x}_{n1} & \widetilde{x}_{n2} & \cdots & \cdots & \widetilde{x}_{nn} \end{bmatrix}, \tag{3}
$$

$$
\widetilde{x}_{ij} = \frac{\widetilde{z}_{ij}}{r} = \left( \frac{Z_{a'_{ij}}}{r}, \frac{Z_{b'_{ij}}}{r}, \frac{Z_{c'_{ij}}}{r}, \frac{Z_{d'_{ij}}}{r}; H_1\!\left(z_{ij}^U\right), H_2\!\left(z_{ij}^U\right) \right), \left( \frac{Z_{e'_{ij}}}{r}, \frac{Z_{f'_{ij}}}{r}, \frac{Z_{g'_{ij}}}{r}, \frac{Z_{h'_{ij}}}{r}; H_1\!\left(z_{ij}^L\right), H_2\!\left(z_{ij}^L\right) \right), \tag{4}
$$

$$
r = max\left( max_{1 \le i \le n} \sum_{j=1}^{n} Z_{d'_{ij}}, max_{1 \le i \le n} \sum_{j=1}^{n} Z_{d_{ij}} \right) \tag{5}
$$

In the next process, total influence fuzzy matrix $(\widetilde{T})$ is generated with the help of Equations (6)–(10) [58]:

$$
\begin{aligned}
X_{\acute{a}} &= \begin{bmatrix} 0 & a'_{12} & \cdots & \cdots & a'_{1n} \\ a'_{21} & 0 & \cdots & \cdots & a'_{2n} \\ \vdots & \vdots & \ddots & \cdots & \cdots \\ \vdots & \vdots & \vdots & \ddots & \vdots \\ a'_{n1} & a'_{n2} & \cdots & \cdots & 0 \end{bmatrix} \\
X_{\acute{h}} &= \begin{bmatrix} 0 & h'_{12} & \cdots & \cdots & h'_{1n} \\ h'_{21} & 0 & \cdots & \cdots & h'_{2n} \\ \vdots & \vdots & \ddots & \cdots & \cdots \\ \vdots & \vdots & \vdots & \ddots & \vdots \\ h'_{n1} & h'_{n2} & \cdots & \cdots & 0 \end{bmatrix}
\end{aligned} \tag{6}
$$

$$
\widetilde{T} = \lim_{k \to \infty} \widetilde{X} + \widetilde{X}^2 + \ldots + \widetilde{X}^k, \tag{7}
$$

$$
\widetilde{T} = \begin{bmatrix} \widetilde{t}_{11} & \widetilde{t}_{12} & \cdots & \cdots & \widetilde{t}_{1n} \\ \widetilde{t}_{21} & \widetilde{t}_{22} & \cdots & \cdots & \widetilde{t}_{2n} \\ \vdots & \vdots & \ddots & \cdots & \cdots \\ \vdots & \vdots & \vdots & \ddots & \vdots \\ \widetilde{t}_{n1} & \widetilde{t}_{n2} & \cdots & \cdots & \widetilde{t}_{nn} \end{bmatrix}, \tag{8}
$$

$$
\widetilde{t}_{ij} = \left( a''_{ij}, b''_{ij}, c''_{ij}, d''_{ij}; H_1\!\left(\widetilde{t}_{ij}^U\right), H_2\!\left(\widetilde{t}_{ij}^U\right) \right), \left( e''_{ij}, f''_{ij}, g''_{ij}, h''_{ij}; H_1\!\left(\widetilde{t}_{ij}^L\right), H_2\!\left(\widetilde{t}_{ij}^L\right) \right), \tag{9}
$$

$$
\left[ a''_{ij} \right] = X_{\acute{a}} \times (I - X_{\acute{a}})^{-1}, \ldots, \left[ h''_{ij} \right] = X_{\acute{h}} \times \left( I - X_{\acute{h}} \right)^{-1}. \tag{10}
$$

The final step is related to the calculation of the influence degrees. Within this framework, the sums of all vector rows $(\widetilde{D}_i)$ and columns $(\widetilde{R}_i)$ are found using Equations (11) and (12):

$$
\widetilde{D}_i = \left[ \sum_{j=1}^{n} \widetilde{t}_{ij} \right]_{n \times 1}, \tag{11}
$$

$$
\widetilde{R}_i = \left[ \sum_{i=1}^{n} \widetilde{t}_{ij} \right]'_{1 \times n}. \tag{12}
$$

Additionally, the value of $\left(\widetilde{D}_i + \widetilde{R}_i\right)$ gives information about the total degree of the influence. When this value is higher, it shows that the factor is closer to the central point. Furthermore, Equations (13)–(16) are used for the calculation of the defuzzified values:

$$Def_T = \frac{\frac{(u_U - l_U) + (\beta_U \times m_{1U} - l_U) + (\alpha_U \times m_{2U} - l_U)}{4} + l_U + \left[ \frac{(u_L - l_L) + (\beta_L \times m_{1L} - l_L) + (\alpha_L \times m_{2L} - l_L)}{4} + l_L \right]}{2}, \quad (13)$$

$$Def_T = T = \left[ t_{ij} \right]_{n \times n}, \; i, j = 1, 2, \ldots, n, \quad (14)$$

$$\widetilde{D}_i^{def} = r = \left[ \sum_{j=1}^{n} t_{ij} \right]_{n \times 1} = (r_i)_{n \times 1} = (r_1, \ldots, r_i, \ldots, r_n), \quad (15)$$

$$\widetilde{R}_i^{def} = y = \left[ \sum_{i=1}^{n} t_{ij} \right]'_{1 \times n} = \left( y_j \right)'_{1 \times n} = (y_1, \ldots, y_i, \ldots, y_n). \quad (16)$$

### 3.2. IT2 Fuzzy VIKOR

VIKOR is also a multi-criteria decision-making method that aims to rank different alternatives according to their significance. In this process, the closeness to the ideal solution is taken into the account. It is also possible to consider this approach using IT2 fuzzy sets. In the first step, expert evaluations are obtained. Then, these evaluations are converted into IT2 fuzzy sets [59]. As such, Table 1 is utilized.

**Table 1.** Linguistic evaluations and fuzzy numbers for the alternatives.

| Linguistic Scales | Interval Type 2 Fuzzy Numbers |
|---|---|
| Very low (VL) | ((0,0,0,0.1;1,1), (0,0,0,0.05;0.9,0.9)) |
| Low (L) | ((0,0.1,0.1,0.3;1,1), (0.05,0.1,0.1,0.2;0.9,0.9)) |
| Medium low (ML) | ((0.1,0.3,0.3,0.5;1,1), (0.2,0.3,0.3,0.4;0.9,0.9)) |
| Medium (M) | ((0.3,0.5,0.5,0.7;1,1), (0.4,0.5,0.5,0.6;0.9,0.9)) |
| Medium high (MH) | ((0.5,0.7,0.7,0.9;1,1), (0.6,0.7,0.7,0.8;0.9,0.9)) |
| High (H) | ((0.7,0.9,0.9,1;1,1), (0.8,0.9,0.9,0.95;0.9,0.9)) |
| Very high (VH) | ((0.9,1,1,1;1,1), (0.95,1,1,1;0.9,0.9)) |

The second step is related to the generation of the fuzzy decision matrix ($X_{ij}$). For this purpose, average values are used, where the details are given in Equations (17) and (18). In these equations, $K$ indicates the number of decision-makers and $X_{ij}$ gives information about the aggregated fuzzy ratings [60,61].

$$X_{ij} = \begin{array}{c} \\ A_1 \\ A_2 \\ A_3 \\ \vdots \\ A_m \end{array} \begin{array}{ccccc} C_1, & C_2, & C_3, & \ldots, & C_n \\ \left[ \begin{array}{ccccc} x_{11} & x_{12} & x_{13} & \ldots & x_{1n} \\ x_{21} & x_{22} & x_{23} & \ldots & x_{2n} \\ x_{31} & x_{32} & x_{33} & \ldots & x_{3n} \\ \vdots & \vdots & \vdots & \ddots & \vdots \\ x_{m1} & x_{m2} & x_{m3} & \ldots & x_{mn} \end{array} \right] \end{array}, \quad (17)$$

$$X_{ij} = \frac{1}{K} \left[ \sum_{e=1}^{n} X_{ij}{}^e \right], \; i = 1, 2, 3, \ldots, m. \quad (18)$$

The third stage of this methodology is related to the defuzzification process. In this step, Equations (19)–(22) are used.

$$
\begin{aligned}
Def\left(x_{ij}\right) = \;& Rank(\widetilde{x}_{ij})_{m\times n} \\
= \;& M_1\!\left(\widetilde{A}_i^U\right) + M_1\!\left(\widetilde{A}_i^L\right) + M_2\!\left(\widetilde{A}_i^U\right) + M_2\!\left(\widetilde{A}_i^L\right) + M_3\!\left(\widetilde{A}_i^U\right) + M_3\!\left(\widetilde{A}_i^L\right) \\
& - \tfrac{1}{4}\left(S_1\!\left(\widetilde{A}_i^U\right) + S_1\!\left(\widetilde{A}_i^L\right) + S_2\!\left(\widetilde{A}_i^U\right) + S_2\!\left(\widetilde{A}_i^L\right) + S_3\!\left(\widetilde{A}_i^U\right) + S_3\!\left(\widetilde{A}_i^L\right)\right. \\
& \left. + S_4\!\left(\widetilde{A}_i^U\right) + S_4\!\left(\widetilde{A}_i^L\right)\right) + H_1\!\left(\widetilde{A}_i^U\right) + H_1\!\left(\widetilde{A}_i^L\right) + H_2\!\left(\widetilde{A}_i^U\right) \\
& + H_2\!\left(\widetilde{A}_i^L\right),
\end{aligned}
\tag{19}
$$

$$
M_p\!\left(\widetilde{A}_i^j\right) = \frac{\left(a_{ip}^j + a_{i(p+1)}^j\right)}{2},
\tag{20}
$$

$$
S_q\!\left(\widetilde{A}_i^j\right) = \sqrt{\frac{1}{2}\sum_{k=q}^{q+1}\left(a_{ik}^j - \frac{1}{2}\sum_{k=q}^{q+1} a_{ik}^j\right)^2},
\tag{21}
$$

$$
S_4\!\left(\widetilde{A}_i^j\right) = \sqrt{\frac{1}{4}\sum_{k=1}^{4}\left(a_{ik}^j - \frac{1}{4}\sum_{k=1}^{4} a_{ik}^j\right)^2}.
\tag{22}
$$

In the fourth step, the best and worst values $(f_j{}^*, f_j{}^-)$ of the matrix are calculated using Equation (23):

$$
f_j^* = \max_i x_{ij} \text{ and } f_j^- = \min_i x_{ij}
\tag{23}
$$

The fifth step involves the calculation of the $S_i$ and $R_i$ values [62], which is detailed in Equations (24) and (25):

$$
S_i = \sum_{i=1}^{n} w_j \frac{\left(\left|f_j^* - x_{ij}\right|\right)}{\left(\left|f_j^* - f_j^-\right|\right)}
\tag{24}
$$

$$
R_i = \max j \left[ w_j \frac{\left(\left|f_j^* - x_{ij}\right|\right)}{\left(\left|f_j^* - f_j^-\right|\right)} \right]
\tag{25}
$$

The final step is related to the computation of $Q_i$ using Equation (26):

$$
Q_i = \frac{v(S_i - S^*)}{(S^- - S^*)} + (1-v)\frac{(R_i - R^*)}{(R^- - R^*)}.
\tag{26}
$$

In this equation, $S^-$ and $R^-$ indicate the maximum values, while the minimum values are given as $S^*$ and $R^*$. The analysis results are checked using two different conditions: the first one is related to the acceptable advantage and while the second one is acceptable stability. The method for calculating these two conditions are demonstrated in Equations (27) and (28):

$$
Q\!\left(A^{(2)}\right) - Q\!\left(A^{(1)}\right) \geq \frac{1}{(j-1)},
\tag{27}
$$

$$
Q\!\left(A^{(M)}\right) - Q\!\left(A^{(1)}\right) < \frac{1}{(j-1)}.
\tag{28}
$$

### 3.3. IT2 Fuzzy TOPSIS

TOPSIS is another multi-criteria decision-making approach. By using this methodology, it is possible to rank different alternatives according to their significance [53]. In the analysis process,

positive and negative ideal solutions ($A^+$, $A^-$). are considered; these are found using Equation (29). In this equation, the weighted values are given as $v_{ij}$.

$$A^+ = \max(v_1, v_2, v_3, \ldots v_n); \; A^- = \min(v_1, v_2, v_3, \ldots v_n) \tag{29}$$

In addition, the distances ($D^+$, $D^-$) are calculated using Equations (30) and (31):

$$D_i{}^+ = \sqrt{\sum_{i=1}^{m} (v_i - A_i{}^+)^2}, \tag{30}$$

$$D_i^- = \sqrt{\sum_{i=1}^{m} (v_i - A_i^-)^2}. \tag{31}$$

In the final stage, the values of the closeness coefficient ($CC_i$) are calculated using Equation (32) with the aim of ranking different alternatives [54]:

$$CC_i = \frac{D_i^-}{D_i^+ + D_i^-}. \tag{32}$$

## 4. A Risk Analysis of Wind Energy Investments in E7 Economies

In this study, the effectiveness of risk management in wind energy investments in E7 countries was examined. In this context, first, the types of systematic risks existing in these investments were identified. In this process, a large literature review was conducted. Subsequently, the determined dimensions and criteria were weighted using the IT2 fuzzy DEMATEL approach. In the last stage of the analysis, the E7 countries were listed in terms of their risk management effectiveness regarding wind energy investments. In order to achieve this goal, the IT2 fuzzy VIKOR model was used. Furthermore, the IT2 fuzzy TOPSIS method was applied as a robustness check.

### 4.1. Determining Criteria and Alternative Countries

During the analysis, first, the systematic risk factors in wind energy investments were determined. In this context, current studies on this subject were examined. As a result of the analysis, 12 different criteria were determined based on 4 different dimensions. Details of the stated factors are shared in Table 2.

**Table 2.** List of the systematic risk factors for wind energy investments.

| Dimensions | Criteria | References |
|---|---|---|
| Macroeconomic factors (D1) | Volatility in the exchange rate (C1)<br>High inflation (C2)<br>Interest rate risk (C3) | [3,5,9]<br>[1,10]<br>[6,11] |
| Socio-political factors (D2) | Political problems (C4)<br>Social conflicts (C5)<br>Legal issues (C6) | [4,5,12]<br>[7,13]<br>[14,23] |
| Market-based factors (D3) | Decrease in market demand (C7)<br>Higher cost (C8)<br>Market competition (C9) | [15,25]<br>[16,26]<br>[17,18] |
| Environmental factors (D4) | Natural disasters (C10)<br>Changing geographical conditions (C11)<br>Unplanned construction (C12) | [18,27]<br>[19,28]<br>[8,28,29] |

As can be seen from Table 2, 12 different systematic risks were identified for wind energy investments; these risks depended on 4 different dimensions. In this process, the first dimension was concered with toward macroeconomic factors. In this context, an increase in the exchange rate will increase the costs incurred by the wind energy investor. The main reason for this is that some materials in wind turbines are imported from abroad. Similar to this issue, the wind energy project can become even more costly when inflation increases. In addition to these factors, the wind power company will have a hard time finding loans for new investments when the interest rate increases. This poses a significant risk for the continuity of the project.

Another dimension that is important in this process is related to sociopolitical factors. First of all, investors are reluctant to make new investments when there are political problems in a country. The reason for this is the risk of changing political power in a country to implement different strategies. These changes may be applications that directly affect wind energy investments. Parallel to this issue, a country that has social disturbances worries investors. Experiencing social conflicts in a country causes an increase in the crime rate, and if these problems cannot be controlled, there is a risk of wind energy investments failing. In addition, wind energy investors will not want to invest in a country where the legal system is constantly and radically changed by political power.

The third dimension identified includes market-based factors. In this context, the decrease in the demand for electrical energy in a country causes the profitability of wind energy companies to decrease. This stated demand may generally decrease in a country where the population is decreasing or where there is an economic recession. In addition, in an environment where taxes increase, there will be serious increases in the cost of items. In this case, it will cause the profitability of wind energy investments to decrease. The last systematic risk type identified in this dimension is the highly competitive environment in the market. In such an environment, wind energy companies will not be flexible in price. This will lead to a decrease in profitability.

Another dimension identified as a systematic risk in wind energy investments is related to environmental factors. In this context, wind energy investments can also be negatively affected in a country where natural disasters occur. For example, natural disasters, such as storms, floods, and earthquakes, may cause serious damage to these projects. In addition, in the case of global warming, serious climate change will occur in the country. This will negatively affect wind energy investments. In addition, building development can have an impact on the performance of wind energy investments in the same region. For example, the effectiveness of wind energy investments is always at risk in an area with unplanned construction.

*4.2. Defining the Significance Levels of Dimensions and Criteria*

The selected criteria, shown in Table 2, were weighted according to their importance. To do this, the IT2 fuzzy DEMATEL approach was used. First, evaluations of three different experts were obtained. These experts each had at least 15 years of experience in the area of risk management in wind energy companies. These people were selected from the upper-level managers and academicians. These experts made their evaluations by considering seven different scales and these evaluations were converted into IT2 fuzzy numbers by considering Table 3.

**Table 3.** Linguistic scales and fuzzy numbers for the criteria and dimensions.

| Linguistic Scales | Interval Type 2 Fuzzy Numbers |
|---|---|
| Very very low (VVL) | ((0,0.1,0.1,0.2;1,1), (0.05,0.1,0.1,0.15;0.9,0.9)) |
| Very low (VL) | ((0.1,0.2,0.2,0.35;1,1), (0.15,0.2,0.2,0.3;0.9,0.9)) |
| Low (L) | ((0.2,0.35,0.35,0.5;1,1), (0.25,0.35,0.35,0.45;0.9,0.9)) |
| Medium (M) | ((0.35,0.5,0.5,0.65;1,1), (0.4,0.5,0.5,0.6;0.9,0.9)) |
| High (H) | ((0.5,0.65,0.65,0.8;1,1), (0.55,0.65,0.65,0.75;0.9,0.9)) |
| Very high (VH) | ((0.65,0.8,0.8,0.9;1,1), (0.7,0.8,0.8,0.85;0.9,0.9)) |
| Very very high (VVH) | ((0.8,0.9,0.9,1;1,1), (0.85,0.9,0.9,0.95;0.9,0.9)) |

Source: Baykasoğlu and Gölcük [63].

After that, the linguistic evaluations of the dimensions and criteria were generated. The details are summarized in the Appendix A (Tables A1–A5). Using these tables, the direct relation matrix was created, which is demonstrated in Table 4.

**Table 4.** Direct relation matrix for the dimensions.

| | D1 | D2 | D3 | D4 |
|---|---|---|---|---|
| D1 | ((0,0,0,0;1,1), (0,0,0,0;0.90,0.90)) | ((0.45,0.60,0.60,0.73;1,1), (0.50,0.60,0.60,0.68;0.90,0.90)) | ((0.70,0.83,0.83,0.93;1,1), (0.75,0.83,0.83,0.88;0.90,0.90)) | ((0.75,0.87,0.87,0.97;1,1), (0.80,0.87,0.87,0.92;0.90,0.90)) |
| D2 | ((0.35,0.50,0.50,0.65;1,1), (0.40,0.50,0.50,0.60;0.90,0.90)) | ((0,0,0,0;1,1), (0,0,0,0;0.90,0.90)) | ((0.60,0.75,0.75,0.87;1,1), (0.65,0.75,0.75,0.82;0.90,0.90)) | ((0.50,0.65,0.65,0.80;1,1), (0.55,0.65,0.65,0.75;0.90,0.90)) |
| D3 | ((0.10,0.22,0.22,0.35;1,1), (0.15,0.22,0.22,0.30;0.90,0.90)) | ((0.10,0.22,0.22,0.35;1,1), (0.15,0.22,0.22,0.30;0.90,0.90)) | ((0,0,0,0;1,1), (0,0,0,0;0.90,0.90)) | ((0.22,0.35,0.35,0.50;1,1), (0.27,0.35,0.35,0.45;0.90,0.90)) |
| D4 | ((0.10,0.20,0.20,0.35;1,1), (0.15,0.20,0.20,0.30;0.90,0.90)) | ((0.17,0.30,0.30,0.45;1,1), (0.22,0.30,0.30,0.40;0.90,0.90)) | ((0.30,0.45,0.45,0.60;1,1), (0.35,0.45,0.45,0.55;0.90,0.90)) | ((0,0,0,0;1,1), (0,0,0,0;0.90,0.90)) |

In the next process, this matrix was normalized, and the new matrix is given in Table 5.

**Table 5.** Normalized direct-relation matrix for the dimensions.

| | D1 | D2 | D3 | D4 |
|---|---|---|---|---|
| D1 | ((0,0,0,0;1,1), (0,0,0,0;0.90,0.90)) | ((0.17,0.23,0.23,0.30;1,1), (0.19,0.23,0.23,0.27;0.90,0.90)) | ((0.27,0.32,0.32,0.35;1,1), (0.28,0.32,0.32,0.34;0.90,0.90) | ((0.28,0.33,0.33,0.37;1,1), (0.30,0.33,0.33,0.35;0.90,0.90)) |
| D2 | ((0.13,0.19,0.19,0.25;1,1), (0.15,0.19,0.19,0.23;0.90,0.90)) | ((0,0,0,0;1,1), (0,0,0,0;0.90,0.90)) | ((0.23,0.28,0.28,0.33;1,1), (0.25,0.28,0.28,0.31;0.90,0.90)) | ((0.19,0.25,0.25,0.32;1,1), (0.21,0.25,0.25,0.30;0.90,0.90)) |
| D3 | ((0.04,0.08,0.08,0.13;1,1), (0.06,0.08,0.08,0.11;0.90,0.90)) | ((0.04,0.08,0.08,0.13;1,1), (0.06,0.08,0.08,0.11;0.90,0.90)) | ((0,0,0,0;1,1), (0,0,0,0;0.90,0.90)) | ((0.08,0.13,0.13,0.19;1,1), (0.10,0.13,0.13,0.17;0.90,0.90)) |
| D4 | ((0.04,0.08,0.08,0.13;1,1), (0.06,0.08,0.08,0.11;0.90,0.90)) | ((0.06,0.11,0.11,0.17;1,1), (0.08,0.11,0.11,0.15;0.90,0.90)) | ((0.11,0.17,0.17,0.23;1,1), (0.13,0.17,0.17,0.21;0.90,0.90)) | ((0,0,0,0;1,1), (0,0,0,0;0.90,0.90)) |

Then, the total relation matrix was calculated, and is given in Table 6.

**Table 6.** Total relation matrix for the dimensions.

| | D1 | D2 | D3 | D4 |
|---|---|---|---|---|
| D1 | ((0.06,0.16,0.16,0.41;1,1), (0.09,0.16,0.16,0.29;0.90,0.90)) | ((0.22,0.37,0.37,0.67;1,1), (0.27,0.37,0.37,0.53;0.90,0.90)) | ((0.37,0.57,0.57,0.92;-;1,1), (0.43,0.57,0.57,0.75;0.90,0.90 | ((0.37,0.55,0.55,0.89;1,1), (0.43,0.55,0.55,0.73;0.90,0.90)) |
| D2 | ((0.16,0.30,0.30,0.56;1,1), (0.20,0.30,0.30,0.44;0.90,0.90)) | ((0.06,0.16,0.16,0.41;1,1), (0.09,0.16,0.16,0.29;0.90,0.90)) | ((0.32,0.50,0.50,0.84;1,1), (0.37,0.50,0.50,0.68;0.90,0.90)) | ((0.27,0.45,0.45,0.79;1,1), (0.33,0.45,0.45,0.64;0.90,0.90)) |
| D3 | ((0.05,0.14,0.14,0.33;1,1), (0.08,0.14,0.14,0.24;0.90,0.90)) | ((0.06,0.15,0.15,0.35;1,1), (0.09,0.15,0.15,0.26;0.90,0.90)) | ((0.04,0.13,0.13,0.34;1,1), (0.07,0.13,0.13,0.24;0.90,0.90)) | ((0.11,0.23,0.23,0.48;1,1), (0.15,0.23,0.23,0.37;0.90,0.90)) |
| D4 | ((0.06,0.15,0.15,0.36;1,1), (0.09,0.15,0.15,0.27;0.90,0.90)) | ((0.08,0.19,0.19,0.41;1,1), (0.12,0.19,0.19,0.31;0.90,0.90)) | ((0.15,0.29,0.29,0.57;1,1), (0.20,0.29,0.29,0.45;0.90,0.90)) | ((0.04,013,0.13,0.36;1,1), (0.07,0.13,0.13,0.26;0.90,0.90)) |

In the next stage, the defuzzification procedure took place. The defuzzified total relation matrix and the weights for the dimensions are illustrated in Table A6. By considering these values, the weights of both the dimensions and criteria were calculated. Table 7 gives the details of these results.

Furthermore, the global weights of the criteria are illustrated in Figure 1.

**Table 7.** Weights of the criteria and dimensions.

| Dimensions | Weights | Criteria | Local Weights |
|---|---|---|---|
| Macroeconomic factors (D1) | 0.266 | Volatility in exchange rate (C1) | 0.348 |
|  |  | High inflation (C2) | 0.325 |
|  |  | Interest rate risk (C3) | 0.326 |
| Socio-political factors (D2) | 0.255 | Political problems (C4) | 0.338 |
|  |  | Social conflicts (C5) | 0.331 |
|  |  | Legal issues (C6) | 0.331 |
| Market-based factors (D3) | 0.239 | Decrease in market demand (C7) | 0.337 |
|  |  | Higher cost (C8) | 0.329 |
|  |  | Market competition (C9) | 0.333 |
| Environmental factors (D4) | 0.240 | Natural disasters (C10) | 0.340 |
|  |  | Changing geographical conditions (C11) | 0.339 |
|  |  | Unplanned construction (C12) | 0.321 |

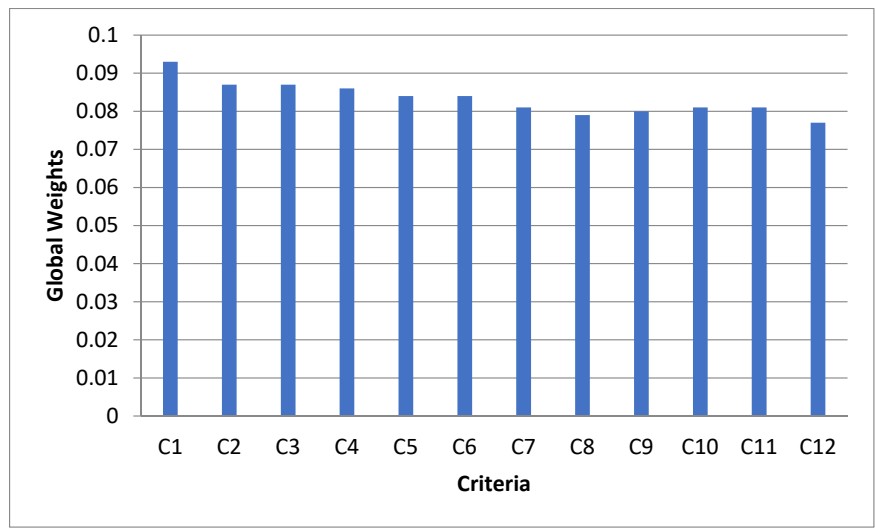

**Figure 1.** Global weights of the criteria.

Table 7 states that the most important dimension was the macroeconomic factors (D1). Socio-political factors (D2) also played a significant role in the systematic risks of wind energy investments. In contrast, the market-based factors (D3) and environmental factors (D4) had lower weights in comparison with others. Regarding the criteria, it was concluded from Figure 1 that volatility in the exchange rate (C1) was the most important item as a systematic risk regarding wind energy investments. Similarly, high inflation (C2), interest rate risk (C3), and political problems (C4) were also other important criteria for this condition. In contrast, social conflicts (C5), legal issues (C6), and changing geographical conditions (C11) played a lower role regarding this situation. However, it was also identified that market competition (C9), higher cost (C8), and unplanned construction (C12) had the least importance.

### 4.3. Evaluations of the E7 Economies Regarding the Risk Management Regarding Wind Energy Investments

In the final stage of the analysis, the E7 countries were ranked in terms of their effectiveness at managing the risk of wind energy investments. In this framework, the IT2 fuzzy VIKOR methodology was used. First, the decision-makers made their evaluations for these seven countries by considering the linguistic scales given in Table 1. These decision-makers each had at least 15 years of experience in this area. They consisted of academicians and top managers in this industry. Hence, it is obvious that they had the necessary background regarding this issue. These evaluations are shown in Table 8.

**Table 8.** Linguistic evaluations of the E7 countries.

| Criteria | China (A1) | | | India (A2) | | | Brazil (A3) | | | Mexico (A4) | | | Russia (A5) | | | Indonesia (A6) | | | Turkey (A7) | | |
|---|---|---|---|---|---|---|---|---|---|---|---|---|---|---|---|---|---|---|---|---|---|
| | DM1 | DM2 | DM3 | DM1 | DM2 | DM3 | DM1 | DM2 | DM3 | DM1 | DM2 | DM3 | DM1 | DM2 | DM3 | DM1 | DM2 | DM3 | DM1 | DM2 | DM3 |
| C1 | H | H | H | H | H | H | H | H | MH | H | H | MH | H | H | VH | H | H | H | VH | H | H |
| C2 | H | MH | MH | VH | VH | H | H | MH | MH | H | MH | MH | M | MH | MH | H | H | VH | M | MH | M |
| C3 | H | MH | MH | VH | H | H | M | MH | MH | M | MH | MH | MH | MH | MH | H | H | H | M | MH | MH |
| C4 | H | H | MH | M | MH | MH | H | VH | H | H | MH | MH | H | MH | MH | H | H | MH | M | MH | MH |
| C5 | H | MH | MH | M | MH | MH | H | MH | MH | MH | H | MH | MH | MH | MH | H | MH | MH | MH | M | MH |
| C6 | M | ML | M | H | MH | MH | M | MH | M | MH | H | MH | M | MH | MH | MH | H | MH | MH | MH | M |
| C7 | H | H | MH | H | MH | MH | H | MH | MH | H | MH | MH | M | MH | MH | H | H | MH | MH | MH | MH |
| C8 | H | MH | MH | M | MH | MH | H | H | MH | H | MH | MH | M | M | MH | H | H | MH | MH | MH | M |
| C9 | H | H | MH | H | H | MH | MH | H | MH | MH | H | MH | M | M | MH | H | H | MH | M | M | MH |
| C10 | H | H | MH | H | MH | MH | H | MH | MH | H | MH | MH | M | MH | MH | H | H | MH | MH | MH | MH |
| C11 | H | MH | MH | M | MH | MH | H | H | MH | H | MH | MH | M | M | MH | H | H | MH | MH | MH | M |
| C12 | H | H | MH | H | H | MH | MH | H | MH | MH | H | MH | M | M | MH | H | H | MH | M | M | MH |

DM: Decision-Makers.

In the next stage, the fuzzy decision matrix was created. The details of this matrix is given in Table A7. After that, the defuzzification process was performed. The final defuzzified decision matrix is given in Table 9.

**Table 9.** Defuzzified decision matrix.

| Criteria | A1 | A2 | A3 | A4 | A5 | A6 | A7 |
|:---:|:---:|:---:|:---:|:---:|:---:|:---:|:---:|
| C1 | 9.03 | 9.03 | 8.64 | 8.64 | 9.25 | 9.03 | 9.25 |
| C2 | 8.26 | 9.47 | 8.26 | 8.26 | 7.47 | 9.25 | 7.07 |
| C3 | 8.26 | 9.25 | 7.47 | 7.47 | 7.87 | 9.03 | 7.47 |
| C4 | 8.64 | 7.47 | 9.25 | 8.26 | 8.26 | 8.64 | 7.47 |
| C5 | 8.26 | 7.47 | 8.26 | 8.26 | 7.87 | 8.26 | 7.47 |
| C6 | 6.27 | 8.26 | 7.07 | 8.26 | 7.47 | 8.26 | 7.47 |
| C7 | 8.64 | 8.26 | 8.26 | 8.26 | 7.47 | 8.64 | 7.87 |
| C8 | 8.26 | 7.47 | 8.64 | 8.26 | 7.07 | 8.64 | 7.47 |
| C9 | 8.64 | 8.64 | 8.26 | 8.26 | 7.07 | 8.64 | 7.07 |
| C10 | 8.64 | 8.26 | 8.26 | 8.26 | 7.47 | 8.64 | 7.87 |
| C11 | 8.26 | 7.47 | 8.64 | 8.26 | 7.07 | 8.64 | 7.47 |
| C12 | 8.64 | 8.64 | 8.26 | 8.26 | 7.07 | 8.64 | 7.07 |

In the final stage, the E7 countries were ranked regarding their effectiveness at the risk management of wind energy investments. The ranking results are demonstrated in Table 10.

**Table 10.** Rankings of the E7 countries' wind investment risk management abilities.

| Countries | $S_i$ | $R_i$ | $Q_i$ | Ranking |
|:---:|:---:|:---:|:---:|:---:|
| China (A1) | 0.220 | 0.059 | 0.021 | 1 |
| India (A2) | 0.486 | 0.086 | 0.618 | 5 |
| Brazil (A3) | 0.256 | 0.087 | 0.461 | 3 |
| Mexico (A4) | 0.394 | 0.087 | 0.563 | 4 |
| Russia (A5) | 0.852 | 0.093 | 0.986 | 6 |
| Indonesia (A6) | 0.191 | 0.084 | 0.372 | 2 |
| Turkey (A7) | 0.871 | 0.093 | 1.000 | 7 |

Table 10 shows that China was the most successful country managing the risk of wind energy investments. Additionally, another successful country in this context was Indonesia. In contrast, it was determined that India, Russia, and Turkey were the least successful countries. The $Q_i$ values were calculated to rank the alternatives using the VIKOR method. Voting by majority rule was used to calculate the final evaluation results. Accordingly, the value of v was defined as 0.5 when consensus was achieved by the experts. However, several strategies of maximum group utility were also used to understand the effects of different voting priorities for the alternatives. Table 11 illustrates the country ranking results using the different strategies of maximum group utility.

**Table 11.** Ranking of countries for different strategies of maximum group utility.

| Countries | v:0 | v:0.1 | v:0.2 | v:0.3 | v:0.4 | v:0.5 | v:0.6 | v:0.7 | v:0.8 | v:0.9 | v:1 |
|:---:|:---:|:---:|:---:|:---:|:---:|:---:|:---:|:---:|:---:|:---:|:---:|
| A1 | 1 | 1 | 1 | 1 | 1 | 1 | 1 | 1 | 1 | 1 | 2 |
| A2 | 3 | 4 | 5 | 5 | 5 | 5 | 5 | 5 | 5 | 5 | 5 |
| A3 | 4 | 3 | 3 | 3 | 3 | 3 | 3 | 3 | 3 | 3 | 3 |
| A4 | 4 | 5 | 4 | 4 | 4 | 4 | 4 | 4 | 4 | 4 | 4 |
| A5 | 6 | 6 | 6 | 6 | 6 | 6 | 6 | 6 | 6 | 6 | 6 |
| A6 | 2 | 2 | 2 | 2 | 2 | 2 | 2 | 2 | 2 | 2 | 1 |
| A7 | 6 | 7 | 7 | 7 | 7 | 7 | 7 | 7 | 7 | 7 | 7 |

According to the results, China (A1) had the best rank for each level of maximum group utility except v:1, while Turkey (A7) has almost the worst rank for the different weights of the decision-making strategy. It is understood that the evaluations of experts were coherent for the different decision-making strategies. The details are demonstrated in Figures 2 and 3.

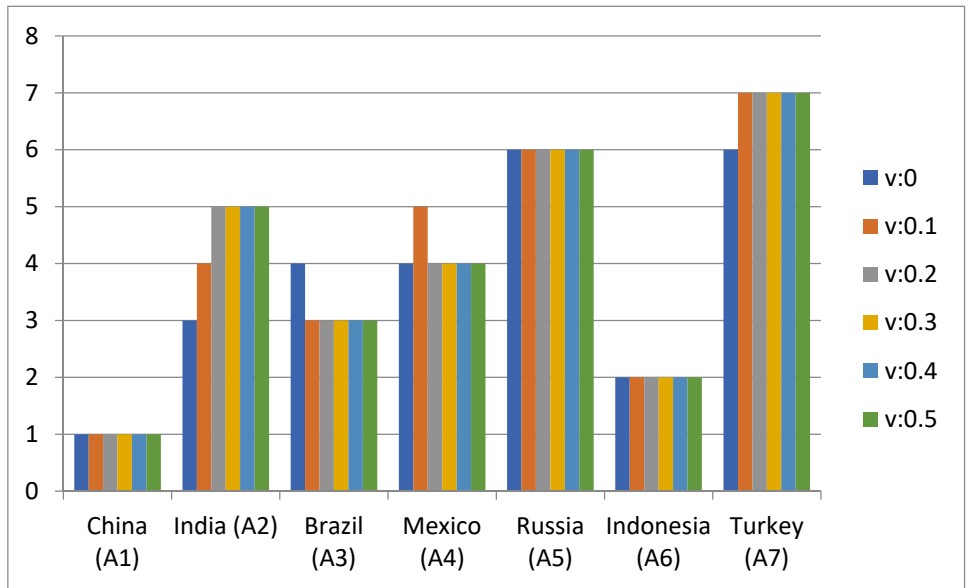

**Figure 2.** Ranking results for (v:0–0.5).

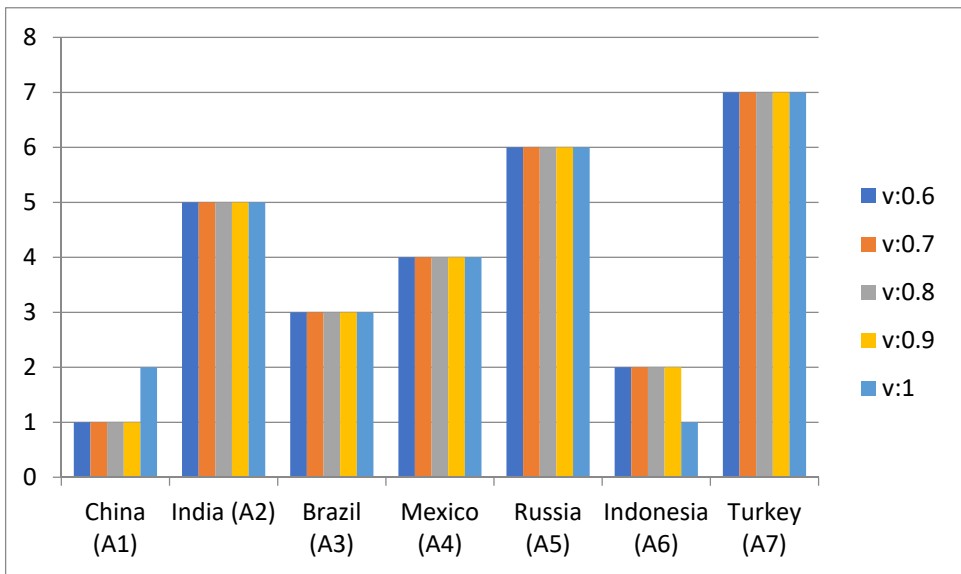

**Figure 3.** Ranking results for (v:0.6–1).

Additionally, the IT2 fuzzy TOPSIS method was applied as a robustness check of the extended VIKOR method. Table 12 illustrates the values of $D^+$, $D^-$, and $CC_i$, as well as the ranking results.

**Table 12.** Ranking results of the countries using IT2 fuzzy TOPSIS.

| Countries | $D^+$ | $D^-$ | $CC_i$ | Ranking |
|-----------|-------|-------|--------|---------|
| A1 | 0.062 | 0.421 | 0.871 | 1 |
| A2 | 0.219 | 0.370 | 0.628 | 2 |
| A3 | 0.228 | 0.316 | 0.581 | 3 |
| A4 | 0.226 | 0.301 | 0.572 | 5 |
| A5 | 0.370 | 0.147 | 0.284 | 6 |
| A6 | 0.227 | 0.312 | 0.579 | 4 |
| A7 | 0.394 | 0.132 | 0.252 | 7 |

Table 12 shows that the ranking results of the IT2 fuzzy TOPSIS were similar to the results of the IT2 fuzzy VIKOR. This is clear evidence that the proposed ranking method was consistent with the comparative analysis results. This situation is also depicted in Figure 4.

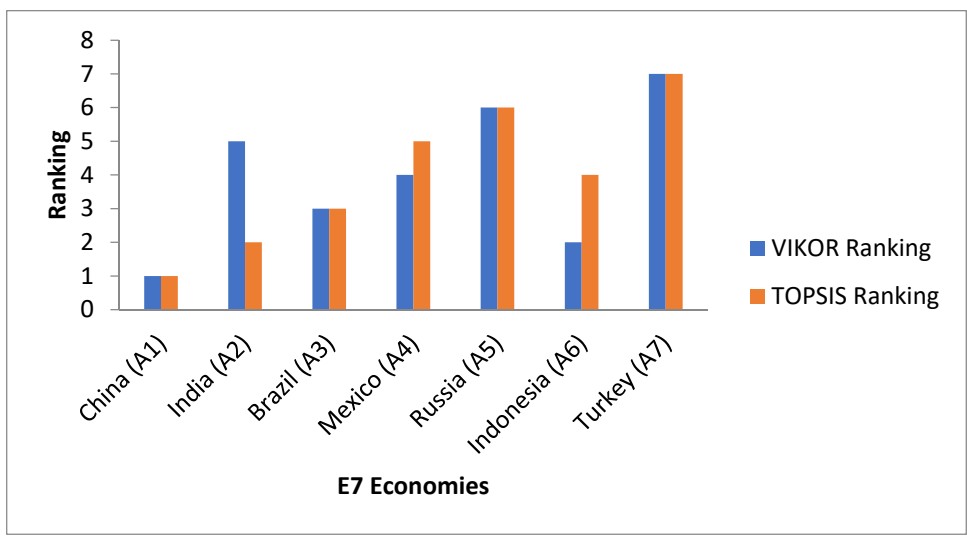

**Figure 4.** Comparative rankings of the interval type 2 (IT2) fuzzy VIKOR and IT2 TOPSIS.

## 5. Discussion and Conclusions

In this study, the systematic risks faced by those making wind energy investments were analyzed. In this context, a literature review was carried out and 12 different types of systematic risks that could have an impact on these investments were identified. It was seen that these risk types belonged to four different dimensions. The analysis process of this study consisted of two different studies. First, an analysis was carried out using the IT2 fuzzy DEMATEL method to determine which of these risk types was more important. In the second part of the analysis, E7 countries were examined regarding their effectiveness at managing systematic risks associated with wind energy investments. In this process, the IT2 fuzzy VIKOR method was preferred for making this comparison between countries. Furthermore, the IT2 fuzzy TOPSIS method was applied as a robustness check.

It was concluded that macroeconomic factors (D1) was the most important dimension. In addition to this issue, it was also identified that socio-political factors (D2) also played an essential role in the risk management of wind energy investments. Another important conclusion was that market-based factors (D3) and environmental factors (D4) had lower weights in comparison with the others. According to the analysis results related to the criteria, it was identified that volatility in the exchange rate (C1) was the most significant systematic risk type regarding wind energy investments. Additionally, high inflation (C2) and interest rate risk (C3) also had important influences on. In contrast, it was also determined that higher costs (C8) and unplanned construction (C12) were the least important risks.

The findings illustrate that wind energy investors should take necessary actions to mainly manage currency exchange risks. For this situation, first of all, a detailed financial analysis should be conducted in the company. As a result of this analysis, it should be understood whether these companies have an open balance sheet position or not. It may be a better idea for these companies to not have an open position. Hence, foreign currency credits should not be preferred to achieve this objective. Another important point is that wind energy companies should adjust the balance of foreign currency assets and liabilities. If there is an open currency position, financial derivatives should be considered to hedge this risk. Reverse currency positions should be taken such that the negative effects of the volatility in currency exchange can be minimized.

Many studies in the literature have found similar results. As an example, Pueyo [3] focused on the leading indicators of renewable energy investments. A comparative analysis was performed between Kenya and Ghana. In this study, a new methodology was proposed using a decision tree approach. It was mainly found that these companies should act to minimize currency exchange rate risks. Similar to this study, Keeley and Matsumoto [5] analyzed the determinants of foreign direct investment in wind and solar energy. For this purpose, developing countries were considered with the help of a semi-structured interview methodology. They concluded that volatility in the currency exchange is one of the most significant risks; as such, these companies should not have an open balance sheet position. In addition, Lei et al. [9] evaluated the different risks in renewable energy projects. In this study, wind energy investments in China were considered. In their study, a Monte Carlo simulation methodology was used. They explained that these companies should not take the loans in foreign currency because an increase in the exchange rate will increase the amount of repayments.

Similar to the currency exchange risk, these companies should also focus on the management of the interest rate risks. For instance, wind energy companies should not take loans that have variable interest rates because any increase in the interest rates harms the effectiveness of these investments. Similarly, interest rate swaps can be very helpful at minimizing interest rate risks for these companies. Another important point is that wind energy companies should also act to manage inflation rate risks. In the case of high inflation, the prices of most of the products have an increasing trend. Therefore, wind energy companies should produce their pricing strategies by considering the current inflation and future inflation predictions. This situation can be very helpful for these companies to have an appropriate price. Otherwise, there is a significant risk that the sales volume may be lower than the increasing costs.

It can be seen that these results were also supported by various academicians in the literature. For instance, Adaramola et al. [1] analyzed wind energy projects. They mainly focused on north-central Nigeria in this study by using a detailed cost analysis. They concluded that as the inflation rate increases, the return on investment decreases such that this risk should be managed effectively. In agreement with this study, Krömer [10] created a model for the risks involved in wind energy production. For this purpose, Monte Carlo simulation methodology was used. It was concluded that the project can become even more costly when inflation increases. In addition to these studies, D'Amico et al. [11] aimed to understand the indicators of wind energy investments by using an indexed semi-Markov chain approach. It was concluded that wind energy companies cannot find new loans when interest rates are high. Moreover, Shammugam et al. [6] studied the supply risks for the development of wind energy in Germany until 2050 by using a regression methodology. It was determined that interest rate risks are one of the most significant risks for these projects.

In the final stage of the analysis process in this study, the E7 countries were ranked in terms of their effectiveness at managing the risks of the wind energy investment process. As a result of the analysis made using IT2 fuzzy VIKOR, it was concluded that China was the most successful country regarding its effectiveness toward managing wind energy investment risks. It was also found that Indonesia and Brazil were also able to successfully manage these risks. In contrast, it was identified that India, Russia, and Turkey were the least capable of managing the associated risks. Additionally, as a result of the IT2 fuzzy TOPSIS analysis, it was found that the proposed ranking method was

consistent with the comparative analysis results. It can be said that the countries that were found to be unsuccessful in this study should mainly focus on minimizing the currency exchange rate, interest rate, and inflation rate risks; otherwise, there is a high risk that wind energy projects in these countries will fail. Therefore, necessary actions should be taken by these countries to manage these risks, such as using financial derivatives, minimizing their open balance sheet position, and not having loans with variable interest rates. Thus, it can be much easier for these countries to have more effective wind energy investment projects. This situation also has a positive influence on the social and economic development of these countries. Based on the results of the analysis conducted in this study, significant strategies can be generated to minimize the risks of wind energy investments. It is believed that these results and strategies can pave the way for both academicians and policymakers. The criteria were identified by making a detailed literature review and weighted for E7 countries. Owing to this condition, it is thought that these strategies can be implemented by the political authorities of these countries to improve renewable energy investment projects.

The main limitation of this study was that it focused only on wind energy projects. Hence, future studies can concentrate on other renewable energy alternatives, such as solar, biomass, geothermal, and hydroelectric energy sources. Evaluating the risk factors in these energy types can also provide coherent results to minimize the risks in these projects. Another important limitation of this study is that only systematic risks were taken into account. Therefore, in a new study, company-based risk factors can also be examined. For instance, liquidity risk, management risk, and portfolio risk can also be analyzed, and new strategies can be produced to manage these risks. In addition, it is also recommended that different methods are used in future studies to make evaluations, such as Multi-Objective Optimization By Ratio Analysis (MOORA) and Electre.

**Author Contributions:** Conceptualization, D.Q. and S.Y.; data curation, D.Q. and G.G.U.; formal analysis, S.Y. and H.D.; funding acquisition, D.Q.; investigation, D.Q., S.Y., H.D., and G.G.U.; methodology, D.Q. and H.D.; resources, S.Y., H.D., and G.G.U.; software, D.Q.; visualization, D.Q.; writing—original draft, D.Q., S.Y., H.D., and G.G.U.; Writing—review and editing, D.Q. and H.D. All authors have read and agreed to the published version of the manuscript.

**Funding:** There was no funding.

**Conflicts of Interest:** The authors declare no conflicts of interest.

## Appendix A

**Table A1.** Linguistic evaluations of the dimensions. D1: Macro-economic factors, D2: Socio-political factors, D3: Market-based factors, D4: Environmental factors.

| | D1 | | D2 | | | | D3 | | | D4 | | |
|---|---|---|---|---|---|---|---|---|---|---|---|---|
| | DM1 | DM2 | DM1 | DM2 | DM3 | DM3 | DM1 | DM2 | DM3 | DM1 | DM2 | DM3 |
| D1 | - | - | - | VH | M | M | VVH | VH | VH | VVH | VH | VVH |
| D2 | M | M | M | - | - | - | VH | VH | H | H | H | H |
| D3 | VVL | VL | L | L | VVL | VL | - | - | - | L | M | VL |
| D4 | VL | VL | VL | L | L | VL | M | M | L | - | - | - |

DM: Decision-Makers.

**Table A2.** Linguistic evaluations of the criteria for dimension 1.

| | C1 | | C2 | | | | C3 | | |
|---|---|---|---|---|---|---|---|---|---|
| | DM1 | DM2 | DM1 | DM2 | DM3 | DM3 | DM1 | DM2 | DM3 |
| C1 | - | - | - | VVH | VH | VH | VH | VVH | VVH |
| C2 | M | H | H | - | - | - | H | VH | VVH |
| C3 | M | M | H | L | L | M | - | - | - |

**Table A3.** Linguistic evaluations of the criteria for dimension 2.

| | C4 | | | C5 | | | C6 | | |
| | DM1 | DM2 | DM1 | DM2 | DM3 | DM3 | DM1 | DM2 | DM3 |
|---|---|---|---|---|---|---|---|---|---|
| C4 | - | - | - | H | M | H | M | M | M |
| C5 | L | L | L | - | - | - | L | VL | L |
| C6 | M | L | M | H | M | H | - | - | - |

**Table A4.** Linguistic evaluations of the criteria for dimension 3.

| | C7 | | | C8 | | | C9 | | |
| | DM1 | DM2 | DM1 | DM2 | DM3 | DM3 | DM1 | DM2 | DM3 |
|---|---|---|---|---|---|---|---|---|---|
| C7 | - | - | - | M | M | H | M | L | H |
| C8 | M | M | M | - | - | - | H | M | M |
| C9 | H | H | M | M | M | L | - | - | - |

**Table A5.** Linguistic evaluations of the criteria for dimension 4.

| | C10 | | | C11 | | | C12 | | |
| | DM1 | DM2 | DM1 | DM2 | DM3 | DM3 | DM1 | DM2 | DM3 |
|---|---|---|---|---|---|---|---|---|---|
| C10 | - | - | - | M | H | M | M | M | M |
| C11 | M | M | L | - | - | - | M | L | M |
| C12 | VL | M | L | L | M | L | - | - | - |

**Table A6.** Defuzzified total relation matrix and the weights for the dimensions.

| | D1 | D2 | D3 | D4 | $R$ | $y$ | $r + y$ | $r - y$ | Weights |
|---|---|---|---|---|---|---|---|---|---|
| D1 | 0.18 | 0.39 | 0.58 | 0.56 | 1.71 | 0.81 | 2.52 | 0.90 | 0.27 |
| D2 | 0.31 | 0.18 | 0.51 | 0.47 | 1.48 | 0.94 | 2.41 | 0.54 | 0.25 |
| D3 | 0.15 | 0.17 | 0.15 | 0.25 | 0.72 | 1.55 | 2.27 | −0.83 | 0.24 |
| D4 | 0.17 | 0.20 | 0.31 | 0.16 | 0.83 | 1.44 | 2.27 | −0.60 | 0.24 |

**Table A7.** Fuzzy decision matrix.

| | A1 | A2 | A3 | A4 |
|---|---|---|---|---|
| C1 | ((0.7,0.9,0.9,1;1,1), (0.8,0.9,0.9,0.95;0.90,0.90)) | ((0.7,0.9,0.9,1;1,1), (0.8,0.9,0.9,0.95;0.90,0.90)) | ((0.63,0.83,0.83,0.97;1,1), 0.73,0.83,0.83,0.9;0.90,0.90) | ((0.63,0.83,0.83,0.97;1,1), 0.73,0.83,0.83,0.9;0.90,0.90) |
| C2 | ((0.57,0.77,0.77,0.93;1,1), (0.67,0.77,0.77,0.85;0.90,0.90)) | ((0.83,0.97,0.97,1;1,1), (0.9,0.97,0.97,0.98;0.90,0.90) | ((0.57,0.77,0.77,0.93;1,1), (0.67,0.77,0.77,0.85;0.90,0.90)) | ((0.57,0.77,0.77,0.93;1,1), (0.67,0.77,0.77,0.85;0.90,0.90)) |
| C3 | ((0.57,0.77,0.77,0.93;1,1), (0.67,0.77,0.77,0.85;0.90,0.90)) | ((0.77,0.93,0.93,1;1,1), (0.85,0.93,0.93,0.97;0.90,0.90)) | ((0.43,0.63,0.63,0.83;1,1), (0.53,0.63,0.63,0.73;0.90,0.90)) | ((0.43,0.63,0.63,0.83;1,1), (0.53,0.63,0.63,0.73;0.90,0.90)) |
| C4 | ((0.63,0.83,0.83,0.97;1,1), 0.73,0.83,0.83,0.9;0.90,0.90) | ((0.43,0.63,0.63,0.83;1,1), (0.53,0.63,0.63,0.73;0.90,0.90)) | ((0.77,0.93,0.93,1;1,1), (0.85,0.93,0.93,0.97;0.90,0.90)) | ((0.57,0.77,0.77,0.93;1,1), (0.67,0.77,0.77,0.85;0.90,0.90)) |
| C5 | ((0.57,0.77,0.77,0.93;1,1), (0.67,0.77,0.77,0.85;0.90,0.90)) | ((0.43,0.63,0.63,0.83;1,1), 0.53,0.63,0.63,0.73;0.90,0.90) | ((0.57,0.77,0.77,0.93;1,1), (0.67,0.77,0.77,0.85;0.90,0.90)) | ((0.57,0.77,0.77,0.93;1,1), (0.67,0.77,0.77,0.85;0.90,0.90)) |
| C6 | ((0.23,0.43,0.43,0.63;1,1), 0.33,0.43,0.43,0.53;0.90,0.90) | ((0.57,0.77,0.77,0.93;1,1), (0.67,0.77,0.77,0.85;0.90,0.90)) | ((0.37,0.57,0.57,0.77;1.00,1.00), (0.47,0.57,0.57,0.67;0.90,0.90)) | ((0.57,0.77,0.77,0.93;1,1), (0.67,0.77,0.77,0.85;0.90,0.90)) |
| C7 | ((0.63,0.83,0.83,0.97;1,1), 0.73,0.83,0.83,0.9;0.90,0.90) | ((0.57,0.77,0.77,0.93;1,1), (0.67,0.77,0.77,0.85;0.90,0.90)) | ((0.57,0.77,0.77,0.93;1,1), (0.67,0.77,0.77,0.85;0.90,0.90)) | ((0.57,0.77,0.77,0.93;1,1), (0.67,0.77,0.77,0.85;0.90,0.90)) |
| C8 | ((0.57,0.77,0.77,0.93;1,1), (0.67,0.77,0.77,0.85;0.90,0.90)) | ((0.43,0.63,0.63,0.83;1,1), 0.53,0.63,0.63,0.73;0.90,0.90) | ((0.63,0.83,0.83,0.97;1,1), 0.73,0.83,0.83,0.9;0.90,0.90) | ((0.57,0.77,0.77,0.93;1,1), (0.67,0.77,0.77,0.85;0.90,0.90)) |
| C9 | ((0.63,0.83,0.83,0.97;1,1), 0.73,0.83,0.83,0.9;0.90,0.90) | ((0.63,0.83,0.83,0.97;1,1), 0.73,0.83,0.83,0.9;0.90,0.90) | ((0.57,0.77,0.77,0.93;1,1), (0.67,0.77,0.77,0.85;0.90,0.90)) | ((0.57,0.77,0.77,0.93;1,1), (0.67,0.77,0.77,0.85;0.90,0.90)) |
| C10 | ((0.63,0.83,0.83,0.97;1,1), 0.73,0.83,0.83,0.9;0.90,0.90) | ((0.57,0.77,0.77,0.93;1,1), (0.67,0.77,0.77,0.85;0.90,0.90)) | ((0.57,0.77,0.77,0.93;1,1), (0.67,0.77,0.77,0.85;0.90,0.90)) | ((0.57,0.77,0.77,0.93;1,1), (0.67,0.77,0.77,0.85;0.90,0.90)) |
| C11 | ((0.57,0.77,0.77,0.93;1,1), (0.67,0.77,0.77,0.85;0.90,0.90)) | ((0.43,0.63,0.63,0.83;1,1), 0.53,0.63,0.63,0.73;0.90,0.90) | ((0.63,0.83,0.83,0.97;1,1), 0.73,0.83,0.83,0.9;0.90,0.90) | ((0.57,0.77,0.77,0.93;1,1), (0.67,0.77,0.77,0.85;0.90,0.90)) |
| C12 | ((0.63,0.83,0.83,0.97;1,1), 0.73,0.83,0.83,0.9;0.90,0.90) | ((0.63,0.83,0.83,0.97;1,1), 0.73,0.83,0.83,0.9;0.90,0.90) | ((0.57,0.77,0.77,0.93;1,1), (0.67,0.77,0.77,0.85;0.90,0.90)) | ((0.57,0.77,0.77,0.93;1,1), (0.67,0.77,0.77,0.85;0.90,0.90)) |

**Table A7.** *Cont.*

| | A5 | A6 | A7 |
|---|---|---|---|
| C1 | ((0.77,0.93,0.93,1;1,1), (0.85,0.93,0.93,0.97;0.90,0.90)) | ((0.7,0.9,0.9,1;1,1), (0.8,0.9,0.9,0.95;0.90,0.90)) | ((0.77,0.93,0.93,1;1,1), (0.85,0.93,0.93,0.97;0.90,0.90)) |
| C2 | ((0.43,0.63,0.63,0.83;1,1), (0.53,0.63,0.63,0.73;0.90,0.90)) | ((0.77,0.93,0.93,1;1,1), (0.85,0.93,0.93,0.97;0.90,0.90)) | ((0.37,0.57,0.57,0.77;1.00,1.00), (0.47,0.57,0.57,0.67;0.90,0.90)) |
| C3 | ((0.5,0.7,0.7,0.9;1,1), (0.6,0.7,0.7,0.8;0.90,0.90)) | ((0.7,0.9,0.9,1;1,1), (0.8,0.9,0.9,0.95;0.90,0.90)) | ((0.43,0.63,0.63,0.83;1,1), (0.53,0.63,0.63,0.73;0.90,0.90)) |
| C4 | ((0.57,0.77,0.77,0.93;1,1), (0.67,0.77,0.77,0.85;0.90,0.90)) | ((0.63,0.83,0.83,0.97;1,1), 0.73,0.83,0.83,0.9;0.90,0.90) | ((0.43,0.63,0.63,0.83;1,1), (0.53,0.63,0.63,0.73;0.90,0.90)) |
| C5 | ((0.5,0.7,0.7,0.9;1,1), (0.6,0.7,0.7,0.8;0.90,0.90)) | ((0.57,0.77,0.77,0.93;1,1), (0.67,0.77,0.77,0.85;0.90,0.90)) | ((0.43,0.63,0.63,0.83;1,1), (0.53,0.63,0.63,0.73;0.90,0.90)) |
| C6 | ((0.43,0.63,0.63,0.83;1,1), (0.53,0.63,0.63,0.73;0.90,0.90)) | ((0.57,0.77,0.77,0.93;1,1), (0.67,0.77,0.77,0.85;0.90,0.90)) | ((0.43,0.63,0.63,0.83;1,1), (0.53,0.63,0.63,0.73;0.90,0.90)) |
| C7 | ((0.43,0.63,0.63,0.83;1,1), (0.53,0.63,0.63,0.73;0.90,0.90)) | ((0.63,0.83,0.83,0.97;1,1), 0.73,0.83,0.83,0.9;0.90,0.90) | ((0.5,0.7,0.7,0.9;1,1), (0.6,0.7,0.7,0.8;0.90,0.90)) |
| C8 | ((0.37,0.57,0.57,0.77;1.00,1.00), (0.47,0.57,0.57,0.67;0.90,0.90)) | ((0.63,0.83,0.83,0.97;1,1), 0.73,0.83,0.83,0.9;0.90,0.90) | ((0.43,0.63,0.63,0.83;1,1), (0.53,0.63,0.63,0.73;0.90,0.90)) |
| C9 | ((0.37,0.57,0.57,0.77;1.00,1.00), (0.47,0.57,0.57,0.67;0.90,0.90)) | ((0.63,0.83,0.83,0.97;1,1), 0.73,0.83,0.83,0.9;0.90,0.90) | ((0.37,0.57,0.57,0.77;1.00,1.00), (0.47,0.57,0.57,0.67;0.90,0.90)) |
| C10 | ((0.43,0.63,0.63,0.83;1,1), (0.53,0.63,0.63,0.73;0.90,0.90)) | ((0.63,0.83,0.83,0.97;1,1), 0.73,0.83,0.83,0.9;0.90,0.90) | ((0.5,0.7,0.7,0.9;1,1), (0.6,0.7,0.7,0.8;0.90,0.90)) |
| C11 | ((0.37,0.57,0.57,0.77;1.00,1.00), (0.47,0.57,0.57,0.67;0.90,0.90)) | ((0.63,0.83,0.83,0.97;1,1), 0.73,0.83,0.83,0.9;0.90,0.90) | ((0.43,0.63,0.63,0.83;1,1), (0.53,0.63,0.63,0.73;0.90,0.90)) |
| C12 | ((0.37,0.57,0.57,0.77;1.00,1.00), (0.47,0.57,0.57,0.67;0.90,0.90)) | ((0.63,0.83,0.83,0.97;1,1), 0.73,0.83,0.83,0.9;0.90,0.90) | ((0.37,0.57,0.57,0.77;1.00,1.00), (0.47,0.57,0.57,0.67;0.90,0.90)) |

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
