# Peer review of "Multi-Faceted Analysis of Systematic Risk-Based Wind Energy Investment Decisions in E7 Economies Using Modified Hybrid Modeling with IT2 Fuzzy Sets"

_energies, doi:10.3390/en13061423_

Round 1

Reviewer 1 Report

I think the paper has potential to be interesting for readers after incorporation of major comments:

Chapter “Introduction”
Some claims were to be substantiated:

e.g.
“E7 countries aim to become developed countries and to achieve this purpose, they try to increase their investment. Wind energy projects can contribute to the purpose of E7 economies. ”
“Vulnerability in macroeconomic conditions can create some systematic risks. For instance, exchange rate risk is an important systematic risk type that wind energy investors have to manage. ”

Some statements are redundant at the Introduction.

Furthermore, some parts of Introduction should be in chapter "theoretical background".
Motivation to solve the problem, specification of the objective and structure of the paper should be part of the introduction chapter.

For example, this part should be moved to theoretical background and then extended : “In addition to these issues, it can also be said that one of the important contributions of this study to literature is methodological. The IT2 fuzzy DEMATEL and IT2 fuzzy VIKOR approaches are considered for the first time in this study on the risks in wind energy investments. "
It would be necessary to compare with other methods in the given area, to identify them and to describe the expected benefits of IT2 fuzzy DEMATEL and IT2 fuzzy VIKOR as a motivation for the implementation of this method. Thus, Chapter 2.2 should be extended.

Results
Risk Factors for Wind Energy Investments are selected in an appropriate manner.
The results should be more commented on the 12 factors examined
Discussion
Discussion should be quite a lot expanded, results should be compared with other studies also regarding used methods. (because one of the contributions of this paper - according authors is fuzzy method).

Author Response

Istanbul, March 5th, 2020

Ms. Cindy Huang
Assistant Editor

Dear Cindy Huang,

According to your last email, please find attached a copy of the revised version of our paper, as a possible publication in the Energies:

Multi-faced analysis of systematic risk-based wind energy investment decisions in E7 economies using a modified hybrid modeling with IT2 fuzzy sets

The comments and suggestions for revision are reflected in the current version of the paper.

A response letter with main changes has been done in this revision together a separated individual response for each reviewer to their comments and a specific reference section to support our answers are attached.

Looking forward to hearing from you.
Thank you.

Yours sincerely,

Serhat Yüksel
Professor of Finance,
İstanbul Medipol University,
The School of Business
e-mail: serhatyuksel@medipol.edu.tr

Response letter to the Reviewers’ Comments on energies-736177

Title of the paper: Multi-faced analysis of systematic risk-based wind energy investment decisions in E7 economies using a modified hybrid modeling with IT2 fuzzy sets

First of all, authors would like to thank all the anonymous Reviewers and the Editors again for their efforts and valuable time to review and improve our paper.
Taking into account their constructive suggestions and comments, the paper has been carefully revised following the referees’ comments.
The main changes in the revised manuscript are:

1. The introduction parts are improved.
2. The methodology is improved.
3. The contribution of the study is underlined appropriately.

We provide below the responses to each referee’s comments.

Reviewer #1

Reviewer’s Comments: Some claims were to be substantiated:

e.g.
“E7 countries aim to become developed countries and to achieve this purpose, they try to increase their investment. Wind energy projects can contribute to the purpose of E7 economies. ”
“Vulnerability in macroeconomic conditions can create some systematic risks. For instance, exchange rate risk is an important systematic risk type that wind energy investors have to manage. ”

Some statements are redundant at the Introduction.

Authors’ answers: The redundant sentences are eliminated from the study. In addition to this issue, similar sentences are combined. The new sentence is given below.

“Vulnerability in the macroeconomic conditions can create some systematic risks for wind energy investors, such as exchange rate risk.”

Reviewer’s Comments: Furthermore, some parts of Introduction should be in chapter "theoretical background".

Motivation to solve the problem, specification of the objective and structure of the paper should be part of the introduction chapter.

For example, this part should be moved to theoretical background and then extended : “In addition to these issues, it can also be said that one of the important contributions of this study to literature is methodological. The IT2 fuzzy DEMATEL and IT2 fuzzy VIKOR approaches are considered for the first time in this study on the risks in wind energy investments. "

It would be necessary to compare with other methods in the given area, to identify them and to describe the expected benefits of IT2 fuzzy DEMATEL and IT2 fuzzy VIKOR as a motivation for the implementation of this method. Thus, Chapter 2.2 should be extended.

Authors’ answers: This section is transferred to the part of Chapter 2.2 according to the reviewer’s comments. This new part is given below.

“2.2. Literature Review on Methodology

It is thought that one of the important contributions of this study to the literature is methodological. IT2 fuzzy DEMATEL and IT2 fuzzy VIKOR approaches are taken into consideration for the first time in this study regarding the risks in wind energy investments. Furthermore, IT2 fuzzy TOPSIS method is also used for robustness check. This situation increases the methodological originality of the study. The biggest advantage of the DEMATEL model over other similar methods is that it also shows causality relationship between the variables. Moreover, group utility maximization and individual regret minimization can be considered in VIKOR approach which is accepted as the main advantage of this method. Furthermore, with the help of IT2 fuzzy methodology, uncertainties in decision making process can be minimized as well.”

Reviewer’s Comments: Results
Risk Factors for Wind Energy Investments are selected in an appropriate manner.
The results should be more commented on the 12 factors examined.

Authors’ answers: Based on the reviewer’s comments, analysis results are detailed. In this framework, we emphasized all 12 different factors. The new version is stated below.

“In the next stage, defuzzification procedure is occurred. Defuzzified total relation matrix and the weights for the dimensions are illustrated in Table A6. By considering these values, the weights of both dimensions and criteria can be calculated. Table 7 explains the details of these results.
Table 7. Weights of the criteria and dimensions
Dimensions Weights Criteria Local Weights
Macroeconomic Factors (D1) 0.266 Volatility in Exchange Rate (C1) 0.348
High Inflation (C2) 0.325
Interest Rate Risk (C3) 0.326
Socio-political Factors (D2) 0.255 Political Problems (C4) 0.338
Social Conflicts (C5) 0.331
Legal Issues (C6) 0.331
Market-based Factors (D3) 0.239 Decrease in Market Demand (C7) 0.337
Higher Cost (C8) 0.329
Market Competition (C9) 0.333
Environmental Factors (D4) 0.240 Natural Disasters (C10) 0.340
Changing Geographical Conditions (C11) 0.339
Unplanned Construction (C12) 0.321
On the other side, the global weights of the criteria are illustrated on Figure 1.
Figure 1. Global Weights of Criteria

Table 7 states that the most important dimension is macroeconomic factors (D1). Socio-political factors (D2) also play a significant role for the systematic risks of wind energy investments. On the other side, market-based factors (D3) and environmental factors (D4) have lower weights in comparison with others. With respect to the criteria, it is concluded form Figure 1 that volatility in exchange rate (C1) is the most important item as a systematic risk in wind energy investments. Similarly, high inflation (C2), interest rate risk (C3) and political problems (C4) are also other important criteria for this condition. In the other hand, it is also defined that social conflicts (C5), legal issues (C6) and changing geographical conditions (C11) play a lower role regarding this situation. However, it is also identified that market competition (C9), higher cost (C8) and unplanned construction (C12) are on the last ranks.”

Reviewer’s Comments: Discussion

Discussion should be quite a lot expanded, results should be compared with other studies also regarding used methods. (because one of the contributions of this paper - according authors is fuzzy method).

Authors’ answers: The discussion part is improved in a detailed manner. Within this scope, results are compared with other studies also regarding used methods. This new part is stated below.

“5. Discussion and Conclusions
In this study, systematic risks existing in wind energy investments are analyzed. In this context, primarily, a literature review was made and 12 different types of systematic risks that could have an impact on these investments were identified. It is seen that these risk types belong to 4 different dimensions. The analysis process of this study consists of two different studies. Firstly, an analysis was carried out with IT2 fuzzy DEMATEL method in order to determine which of these risk types are more important. On the other hand, in the second part of the analysis part of the study, E7 countries have been examined for their effectiveness in the management of systematic risks in wind energy investments. In this process, IT2 fuzzy VIKOR method was preferred to make this comparison between countries. Furthermore, IT2 fuzzy TOPSIS method is applied for robustness check as well.
It is concluded that macroeconomic factors (D1) is the most important dimension. In addition to this issue, it is also identified that socio-political factors (D2) also play an essential role for the systematic risks of wind energy investments. Another important conclusion is that market-based factors (D3) and environmental factors (D4) have lower weights in comparison with others. According to the analysis results related to the criteria, it is identified that volatility in exchange rate (C1) is the most significant systematic risk type for wind energy investments. Additionally, high inflation (C2) and interest rate risk (C3) have also important influence on this condition. On the other side, it is also determined that higher cost (C8) and unplanned construction (C12) take place on the last ranks.
The findings illustrate that wind energy investors should take necessary actions mainly to manage currency exchange risk. For this situation, first of all, a detailed financial analysis should be conducted in the company. As a result of this analysis, it should be understood whether these companies have open balance sheet position or not. It can be a better idea for these companies not to have open position. Hence, foreign currency credits should not be preferred to achieve this objective. Another important point is that wind energy companies should adjust the balance of foreign currency assets and liabilities. If there is an open currency position, financial derivatives should be taken into account to hedge this risk. Reverse positions should be taken in currency so that the negative effects of the volatility in currency exchange can be minimized.
Lots of the studies in the literature underlined the importance of the similar results. As an example, Pueyo [3] focused on the leading indicators of renewable energy investments. A comparative analysis was performed between Kenya and Ghana. In this study, new methodology is proposed by using decision tree approach. It is mainly defined that these companies should take actions to minimize currency exchange rate risks. Similar to this study, Keeley and Matsumoto [5] made an analysis to understand the determinants of foreign direct investment in wind and solar energy. For this purpose, developing countries are taken into account with the help of semi-structured interview methodology. They reached a conclusion that volatility in currency exchange is one of the most significant risks so that these companies should not have open balance sheet position. In addition to them, Lei et al. [9] evaluated the different risks in renewable energy projects. In this study, wind energy investments in China are taken into consideration. In the analysis process of this study, Monte Carlo simulation methodology is taken into consideration. They mainly discussed that these companies should not take the loans in foreign currency because the increase in the exchange rate will increase the amount of repayments.
Similar to the currency exchange risk, these companies should also focus on the management of the interest rate risk. For instance, wind energy companies should not take the loans which have variable interest rate. Otherwise, any increase in the interest rate have a negative impact on the effectiveness of these investments. Similarly, interest rate swaps can be very helpful to minimize interest rate risks for these companies. Another important point is that wind energy companies should also take actions to manage inflation rate risks. In case of high inflation, the prices of most of the products have an increasing trend. Therefore, wind energy companies should define their pricing strategies by considering current inflation and future inflation predictions. This situation can be very helpful for these companies to have an appropriate price. Otherwise, there is a significant risk that sales volume can be lower than the increasing costs.
It can be seen that these results were also supported by various academicians in the literature. For instance, Adaramola et al. [1] made an assessment regarding wind energy projects. They mainly focused on north-central Nigeria in this study by using a detailed cost analysis. They reached a conclusion that as inflation rate increases, return on investment decreases so that this risk should be managed effectively. Parallel to this study, Krömer [10] tried to create a model for the risks in wind energy production. For this purpose, Monte Carlo simulation methodology is taken into account. It is concluded that the project can become even more costly when inflation increases. In addition to these studies, D’Amico et al. [11] aimed to understand the indicators of wind energy investments by considering indexed semi-Markov chain approach. It is mainly underlined that wind energy companies cannot find new loans in case of the fact that interest rates are high. Moreover, Shammugam et al. [6] made a study for the supply risks for the development of wind energy in Germany until 2050 by using regression methodology. It is determined that interest rate risk is one of the most significant risks for these projects.
In the final stage of the analysis process in this study, E7 countries are ranked regarding the effectiveness of the risk management in the wind energy investment process. As a result of the analysis made with IT2 fuzzy VIKOR, it is concluded that China is the most successful country with respect to the effectiveness in the risk management of the wind energy investments. It is also defined that Indonesia and Brazil are also other countries which are very successful to manage these risks. On the other side, it is also identified that India, Russia and Turkey are the countries on the last rank. Additionally, as a result of IT2 fuzzy TOPSIS analysis, it is also defined that the proposed ranking method is consistent with the comparative analysis results. It can be said that the countries, which are found as unsuccessful in this study, should mainly focus on minimizing currency exchange rate, interest rate and inflation rate risks. Otherwise, there is a high risk that wind energy projects in these countries fail. Therefore, necessary actions should be taken by these countries to manage these risks, such as using financial derivatives, minimizing open balance sheet position and not having loans with variable interest rates. Thus, it can be much easier for these countries to have more effective wind energy investment projects. This situation has also positive influence on the social and economic development of these countries. As a result of the analysis conducted in this study, significant strategies can be generated to minimize the risks of wind energy investments. It is believed that these results and strategies can pave the way for both academicians and policy makers. The criteria are identified by making a detailed literature review and weighted for E7 countries. Owing to this condition, it is thought that these strategies can be implemented by the political authorities of these countries with the aim of improving renewable energy investment projects.
The main limitation of this study is focusing on only wind energy projects. Hence, the future studies can concentrate on other renewable energy alternatives, such as solar, biomass, geothermal and hydroelectric. Evaluating the risk factors in these energy types can also provide coherent results to minimize the risks in these projects. Another important limitation of this study is that only systematic risks are taken into account. Therefore, in a new study, company-based risk factors can also be examined. For instance, liquidity risk, management risk and portfolio risk can also be analyzed, and new strategies can be prevented to manage these risks. In addition to them, it is also recommended that different methods can be preferred in the future studies to make evaluations, such as MOORA and ELECTRE.”

Reviewer 2 Report

The study analyzes and compares the risks of investment in wind energy in the various E7 countries. First, a multiple of risk types are weighted for significance and their mutual influences determined, and secondly, expert evaluation economics and other factors of the various countries are ranked in view of successful management of those risks. Suitable methods available for such study are applied. The structure of the paper is systematic and good. After an extensive problem analysis both about socio-economic aspects and physical environmental factors, a description of the analysis process is given after which results and conclusions. The description of the analysis process the study followed is clear, although very concise. Readers interested in details shall have to consult various references. The uncertainty expressed in the IT2 FS values picked by the experts vanishes by the defuzzification step. Because the article does not show any graphics for a reader to get an impression of the uncertainty takes an effort. Can you show that this defuzzification does not affect the ranking? (Wang et al., 2019 - ref.47 follow a different way). What is exactly the gain applying IT2 FS instead of T1 FS? Will the ranking become different? It is not made clear from where the three DM's in Table 8 originate that have the overview of how the twelve criteria shall be judged in the seven different countries. What is their authority?
What are you going to do with the study? Is it just to demonstrate what one can do with IT2 FS and the MCDM models, or are you going to offer the results to a representative in the E7 country contact body?
As a minor comment it may be good for a general reader to state somewhere at the start what are the E7 countries and why you selected those.
You explain the symbols well, only DM1-3 Decision Makers 1-3 in Table 8 and the Annex are not explained (At least I couldn't find it).

Author Response

Istanbul, March 5th, 2020

Ms. Cindy Huang

Assistant Editor

Dear Cindy Huang,

According to your last email, please find attached a copy of the revised version of our paper, as a possible publication in the Energies:

Multi-faced analysis of systematic risk-based wind energy investment decisions in E7 economies using a modified hybrid modeling with IT2 fuzzy sets

The comments and suggestions for revision are reflected in the current version of the paper.

A response letter with main changes has been done in this revision together a separated individual response for each reviewer to their comments and a specific reference section to support our answers are attached.

Looking forward to hearing from you.

Thank you.

Yours sincerely,

Serhat Yüksel

Professor of Finance,

İstanbul Medipol University,

The School of Business

e-mail: serhatyuksel@medipol.edu.tr

Response letter to the Reviewers Comments on energies-736177

Title of the paper: Multi-faced analysis of systematic risk-based wind energy investment decisions in E7 economies using a modified hybrid modeling with IT2 fuzzy sets

First of all, authors would like to thank all the anonymous Reviewers and the Editors again for their efforts and valuable time to review and improve our paper.

Taking into account their constructive suggestions and comments, the paper has been carefully revised following the referees’ comments.

The main changes in the revised manuscript are:

  1. The introduction parts are improved.
  2. The methodology is improved.
  3. The contribution of the study is underlined appropriately.

We provide below the responses to each referee’s comments.

Reviewer #2

Reviewer’s Comments: The uncertainty expressed in the IT2 FS values picked by the experts vanishes by the defuzzification step. Because the article does not show any graphics for a reader to get an impression of the uncertainty takes an effort. Can you show that this defuzzification does not affect the ranking? (Wang et al., 2019 - ref.47 follow a different way).

Authors’ answers:

(1) The formula of the defuzzification process for DEMATEL has been detailed. These details are given below.

“Additionally, the value of  gives information about the total degree of the influence. When this value is higher, it shows that the factor is closer to the central point. On the other side, equations (13)-(16) are taken into consideration for the calculation of the defuzzified values.

                       (13)

 ,                                                      (14)

                                        (15)

                                      (16)”

(2) Additionally, for VIKOR, the defuzzified q values between 0-1 for different consensus values were calculated and the results were compared. The details are given below.

“Table 10 identifies that China is the most successful country with respect to the effectiveness in the risk management of the wind energy investments. Additionally, another successful country in this context is Indonesia. On the other hand, it is also determined that India, Russia and Turkey are the countries on the last rank. Qi values are calculated to rank the alternatives with VIKOR method. Voting by majority rule is considered to calculate the final evaluation results. Accordingly, the value of V is defined as 0.5 that defines the consensus in decision making process by the experts. However, several strategies of maximum group utility are also considered to understand the effects of different voting priorities for the alternatives. Table 11 illustrates the ranking results of alternatives by the different strategies of maximum group utility.

Table 11. Ranking of alternatives for different strategies of maximum group utility

Alternatives

v:0

v:0.1

v:0.2

v:0.3

v:0.4

v:0.5

v:0.6

v:0.7

v:0.8

v:0.9

v:1

A1

1

1

1

1

1

1

1

1

1

1

2

A2

3

4

5

5

5

5

5

5

5

5

5

A3

4

3

3

3

3

3

3

3

3

3

3

A4

4

5

4

4

4

4

4

4

4

4

4

A5

6

6

6

6

6

6

6

6

6

6

6

A6

2

2

2

2

2

2

2

2

2

2

1

A7

6

7

7

7

7

7

7

7

7

7

7

According to the results, alternative 1 has the best rank for each level of maximum group utility except v:1 while A7 has almost the worst seat for the different weights of decision making strategy. It is understood that the evaluations of experts are coherent for the different decision making strategies. The details are demonstrated on Figure 2 and 3.

Figure 2. Ranking Results for (v:0-0.5)

Figure 3. Ranking Results for (v:0.6-1)

(3) A new analysis is performed with IT2 fuzzy TOPSIS to check the consistency of ranking rankings for robustness check. Ranking consistencies are compared. The details are given as following.

“Additionally, IT2 fuzzy TOPSIS method is applied for robustness check of the extended VIKOR method. Table 12 illustrates the values of D+, D- and CCİ as well as ranking results.

Table 12. Ranking results of alternatives with IT2 fuzzy TOPSIS

Alternatives

D+

D−

CCi

Ranking

A1

0.062

0.421

0.871

1

A2

0.219

0.370

0.628

2

A3

0.228

0.316

0.581

3

A4

0.226

0.301

0.572

5

A5

0.370

0.147

0.284

6

A6

0.227

0.312

0.579

4

A7

0.394

0.132

0.252

7

Table 12 shows that the ranking results of IT2 fuzzy TOPSIS are similar with the results of IT2 fuzzy VIKOR. This is a clear evidence that the proposed ranking method is consistent with the comparative analysis results.  This situation is also depicted in Figure 4.

Figure 4. Comparative Rankings of IT2 Fuzzy VIKOR and IT2 TOPSIS

Reviewer’s Comments: What is exactly the gain applying IT2 FS instead of T1 FS? Will the ranking become different?

Authors’ answers: This comment is satisfied by explaining the gains of IT2 fuzzy sets more than IT1 fuzzy sets in the introduction part. Moreover, necessary studies are cited in this part as well. This new paragraph is given below.

“In the analysis in this process, IT2 fuzzy VIKOR model will be taken into account. Moreover, IT2 fuzzy TOPSIS method is also considered for robustness check of the extended VIKOR method. According to the results of the analysis, it will be revealed which countries are more successful or fail to manage these risks. The main advantage of considering IT2 fuzzy sets in comparison with IT1 fuzzy sets is that it handles uncertainties better. In addition to this situation, many control designs can also be implemented in IT2 fuzzy sets [21]. Moreover, another important advantage of IT2 fuzzy sets is its relatively high flexibility and robustness by comparing with IT1 fuzzy logic [22].”

Reviewer’s Comments: It is not made clear from where the three DM's in Table 8 originate that have the overview of how the twelve criteria shall be judged in the seven different countries. What is their authority?

Authors’ answers: According to the reviewer’s comments, the details of the decision makers are explained on the part of “4.3. Evaluations E7 Economies Regarding the Risk Management in Wind Energy Investments”. This new paragraph is given below.

“In the final stage of the analysis, E7 countries are ranked regarding the effectiveness of the risk management of wind energy investments. In this framework, IT2 fuzzy VIKOR methodology is taken into account. Firstly, there decision makers made their evaluations for these seven countries by considering the linguistic scales given in Table 1. These decision makers have at least 15-year experience in this area. They consist of academicians and top managers in this industry. Hence, it is obvious that they have necessary background in this issue. These evaluations are shown in Table 8.”

Reviewer’s Comments: What are you going to do with the study? Is it just to demonstrate what one can do with IT2 FS and the MCDM models, or are you going to offer the results to a representative in the E7 country contact body?

Authors’ answers: Strategies will be produced from the analyzes obtained. Therefore, these issues will guide both academics and policy makers. The criteria obtained are designed and weighted for E7 countries based on the literature. It was therefore concluded that these proposals would be beneficial when considered by countries. This situation is explained in the part of “5. Discussion and Conclusions”. These new sentences are demonstrated as following.

“In the final stage of the analysis process in this study, E7 countries are ranked regarding the effectiveness of the risk management in the wind energy investment process. As a result of the analysis made with IT2 fuzzy VIKOR, it is concluded that China is the most successful country with respect to the effectiveness in the risk management of the wind energy investments. It is also defined that Indonesia and Brazil are also other countries which are very successful to manage these risks. On the other side, it is also identified that India, Russia and Turkey are the countries on the last rank. Additionally, as a result of IT2 fuzzy TOPSIS analysis, it is also defined that the proposed ranking method is consistent with the comparative analysis results.  It can be said that the countries, which are found as unsuccessful in this study, should mainly focus on minimizing currency exchange rate, interest rate and inflation rate risks. Otherwise, there is a high risk that wind energy projects in these countries fail. Therefore, necessary actions should be taken by these countries to manage these risks, such as using financial derivatives, minimizing open balance sheet position and not having loans with variable interest rates. Thus, it can be much easier for these countries to have more effective wind energy investment projects. This situation has also positive influence on the social and economic development of these countries. As a result of the analysis conducted in this study, significant strategies can be generated to minimize the risks of wind energy investments. It is believed that these results and strategies can pave the way for both academicians and policy makers. The criteria are identified by making a detailed literature review and weighted for E7 countries. Owing to this condition, it is thought that these strategies can be implemented by the political authorities of these countries with the aim of improving renewable energy investment projects.”

Reviewer’s Comments: As a minor comment it may be good for a general reader to state somewhere at the start what are the E7 countries and why you selected those.

Authors’ answers: Based on the reviewer’s comments, the reason of selecting E7 economies are explained in the introduction part. These new sentences are given below.

“In this study, systematic risks in wind energy investments are analyzed. Within this framework, firstly, as a result of the literature review, 12 different risk types are determined for wind energy investments which are connected to 4 groups. After that, an investigation is carried out with IT2 fuzzy DEMATEL method in order to understand which of these risks are prioritized. Therefore, it will be possible to find the answer to the question of which systematic risks are faced mostly by the wind energy investors.  Following these analyzes, E7 countries (Brazil, China, India, Indonesia, Mexico, Russia, Turkey) are ranked for their risk situations in wind energy investments. These countries have developing economies, but they aim to reach the status of developed economies. For this purpose, these countries try to generate many significant strategies in various areas. The main problem in this process is that these countries can sometimes take high risks so as to achieve this objective [20]. Thus, it is thought that the risk analysis studies are very helpful for these countries to reach sustainable economic improvement.”

Reviewer’s Comments: You explain the symbols well, only DM1-3 Decision Makers 1-3 in Table 8 and the Annex are not explained (At least I couldn't find it).

Authors’ answers: The term “DM” is explained as “decision makers” in all tables in which this term is used.

Round 2

Reviewer 1 Report

I can recommend this improved manuscript for publication.